# Neural Concept Formation in Knowledge Graphs

**Agnieszka Dobrowolska**                                        AGA.DOBROWOLSKA.16@UCL.AC.UK
*University College London, London, United Kingdom*

**Antonio Vergari**                                                    AVER@CS.UCLA.EDU
*University of California Los Angeles, Los Angeles, United States*

**Pasquale Minervini**                                          P.MINERVINI@UCL.AC.UK
*University College London, London, United Kingdom*

## Abstract

In this work, we investigate how to learn novel concepts in Knowledge Graphs (KGs) in a principled way, and how to effectively exploit them to produce more accurate neural link prediction models. Specifically, we show how concept membership relationships learned via unsupervised clustering of entities can be reified and used to augment a KG. In a thorough set of experiments we confirm that neural link predictors trained on these augmented KGs, or in a joint Expectation-Maximization iterative scheme, can generalize better and produce more accurate predictions for infrequent relationships. For instance, our method yields relative improvements of up to 8.6% MRR on WN18RR for rare predicates, and up to 82% in small-data regimes, where the model has access to just a small subset of the training triples. Furthermore, our proposed models are able to learn meaningful concepts.

## 1. Introduction

One of the most remarkable aspects of human intelligence is arguably the capacity to abstract and summarize knowledge into *concepts*. It is believed to play a central role in allowing humans to learn quickly from few examples [Lake et al., 2015] and to robustly generalize to unseen data [Pothos and Chater, 2002, Lakoff and Johnson, 1980, Rosch et al., 1976]. It is no wonder that many machine learning and knowledge representation methods have tried to "reverse-engineer" how humans learn concepts [Tenenbaum, 2018, Hassabis et al., 2017] in order to automate reasoning as well as knowledge base construction [Kok and Domingos, 2007, Kemp et al., 2006, Xu et al., 2006]. Among the most prominent knowledge representation formalisms, there are *Knowledge Graphs* (KGs) – graph-structured knowledge bases where knowledge about the world is encoded in the form of relationships between entities. KGs encode facts about entities and the relationships between them (edges) as *subject-predicate-object* triples, each denoting a relationship of type *predicate* between the *subject* and *object* of the triple. A fundamental task in the construction of KGs is *link prediction*, which consists of identifying missing edges between entities in the KG.

*Neural link predictors* are a class of link prediction models achieving state-of-the-art results on several link prediction benchmarks while being able to scale to very large KGs. Neural link predictors learn embedding representations for each entity and relation in the KG via back-propagation [Nickel et al., 2016]. However, neural link predictors are known not to be accurate in the presence of sparse KGs, i.e., when entities appear only in few

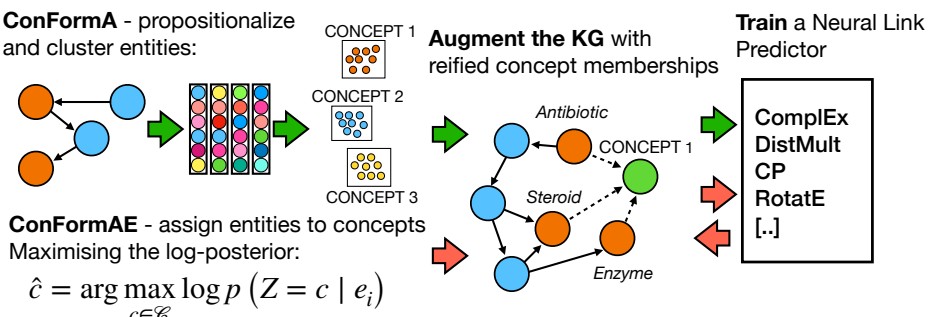

Figure 1: High-level visualization of CONFORMA and CONFORMAE: entities are propositionalized and clustered in *concepts*, and concept memberships are added to the KG as triples. In CONFORMAE, concept memberships are iteratively revised to maximize the likelihood of the data.

triples [Pujara et al., 2017], also referred to as the *cold-start problem*, and may not be able to learn patterns involving *groups of entities* [Evans and Grefenstette, 2018].

In this work, we propose to *learn concepts* in neural link predictors as a principled way to elicit discrete latent information that can alleviate the generalization issues of existing models. Moreover, learning new concepts and turning them into entities can help to automate the construction of KGs. Specifically, we make the following contributions. First, we formalize concept learning as an unsupervised clustering step over entities in a KG. We do this first by *reifying* concept membership relationships into KG facts and by incorporating them in the KG, in a process called *KG augmentation*, akin to Gad-Elrab et al. [2020]. Then, we demonstrate that training out-of-the-box neural link predictors on these augmented KGs improves their accuracy. Secondly, we introduce a single, principled probabilistic framework for jointly learning concept memberships and neural link prediction models at once, by maximizing the likelihood of the KG triples via an Expectation-Maximization scheme. Lastly, we execute a rigorous empirical evaluation on several real-world KG benchmarks, showing that both of our approaches, named CONFORMA and CONFORMAE, are capable of learning semantically meaningful concepts. We observe that explicitly augmenting the KG with the newly-learned concepts can improve generalization over rare predicates by up to 8.6% in terms of Mean Reciprocal Rank (MRR) on WN18RR and 2.1% on FB15k-237. Furthermore, we perform a sparsification analysis where neural link predictors are trained on only a small subset of the training set, showing that CONFORMA and CONFORMAE can achieve up to 82% relative improvement on WN18RR and 21% on FB15k-237 when trained on only 5% of the data. This highlights the potential of our approach for addressing the cold-start problem in sparse KGs.

## 2. Background

Let a KG $\mathcal{G}$ be represented as a set of $N$ triples, i.e., $\mathcal{G} = \{\langle s, r, o \rangle_i\}_{i=1}^{N} \subseteq \mathcal{E} \times \mathcal{R} \times \mathcal{E}$ where $\mathcal{E} = \{e_i\}_{i=1}^{N_e}$ is the set of subject ($s$) and object ($o$) entities, and $\mathcal{R} = \{r_i\}_{i=1}^{N_r}$ the set of relation types ($r$). Neural link predictors can be framed as learning a $k$-dimensional

---

**Algorithm 1** ConFormA($\mathcal{G}$, $N_c$, $n$)

---

1: **Input:** KG $\mathcal{G}$, no. of clusters $N_c$, no. of epochs $n$
2: **Output:** Parameters $\boldsymbol{\Theta}$, cluster memberships $\mathcal{S} = \{S_c \mid c \in \mathcal{C}\}$
3: $\mathbf{P} \leftarrow$ propositionalization($\mathcal{G}$)                          ▷ E.g., random paths
4: $\mathcal{S} \leftarrow$ Clustering($\mathbf{P}, N_c$)                          ▷ E.g., spectral clustering
5: $\mathcal{G}' \leftarrow \mathcal{G} \cup \{\langle e_i, \text{ISA}, c_j \rangle \mid e_i \in S_c, c \in \mathcal{C}\}$          ▷ Create an augmented KG
6: $\boldsymbol{\Theta} \leftarrow$ init()
7: **for** $n$ epochs **do**          ▷ Train the parameters $\boldsymbol{\Theta}$ of a neural link predictor on $\mathcal{G}'$
8:     $\boldsymbol{\Theta} \leftarrow$ train($\mathcal{G}', \boldsymbol{\Theta}$)
    **return** $\boldsymbol{\Theta}, \mathcal{S}$

---

representation, i.e., an *embedding* vector $\mathbf{e} \in \mathbb{C}^k$, for all entities in $\mathcal{E}$ appearing in $\mathcal{G}$. Given a triple $\langle s, r, o \rangle \in \mathcal{E} \times \mathcal{R} \times \mathcal{E}$, a neural link predictor defines a scoring function $\phi_r : \mathbb{C}^k \times \mathbb{C}^k \mapsto \mathbb{R}$ that, given the embedding representations $\mathbf{e}_s \in \mathbb{C}^k$ and $\mathbf{e}_o \in \mathbb{C}^k$ of the subject $s$ and the object $o$ of the triple, returns the score $\phi_r(\mathbf{e}_s, \mathbf{e}_o) \in \mathbb{R}$ that $s$ and $o$ are related by the relation type $r$. The scoring function $\phi_r$ implicitly defines a probability distribution $p$ over triples. Let $x_{sro}$ be a binary random variable denoting the existence of the triple $\langle s, r, o \rangle$, then we have that the probability of observing the triple is proportional to its score, i.e., $p(x_{sro}) \propto \phi_r(\mathbf{e}_s, \mathbf{e}_o)$ [Nickel et al., 2016]. Examples of neural link predictors are TransE [Bordes et al., 2013], DistMult [Yang et al., 2015b], and ComplEx [Trouillon et al., 2016]. In this work, we adopt the latter two as baselines, as they achieve state-of-the-art performance on a number of benchmarks when trained with care [Ruffinelli et al., 2020].

Let $\mathbf{E} \in \mathbb{C}^{N_e \times k}$ and $\mathbf{W} \in \mathbb{C}^{N_r \times k}$ be the relation and entity embeddings. Then, learning a neural link predictor from data consists of finding the best set of parameters $\boldsymbol{\Theta} = \{\mathbf{E}, \mathbf{W}\}$ that solve the optimization problem $\arg\min_{\boldsymbol{\Theta}} \mathcal{L}(\mathcal{G}; \boldsymbol{\Theta}) + \lambda\Omega(\boldsymbol{\Theta})$, where $\mathcal{L}(\mathcal{G}; \boldsymbol{\Theta})$ denotes a model-specific loss, typically proportional to the negative log-likelihood of the triples [Trouillon et al., 2016], and $\Omega(\cdot)$ is a regularization term, such as the $L_2$ or the nuclear weighted 3-norm [N3, Lacroix et al., 2018], whose weight is determined by a coefficient $\lambda \geq 0$. In the following section, we discuss how this learning loop for neural link predictors can be extended to learn concepts in a KG, *while treating the link predictor model as a black-box.*

## 3. ConFormA: Learning Concepts by Augmenting Knowledge Graphs

Given a KG $\mathcal{G}$, we define *concept learning* as identifying sets of entities $S_1, \ldots, S_{N_c} \subseteq \mathcal{E}$ that are semantically related and can be *abstracted* into concepts $c_1, \ldots, c_{N_c} \in \mathcal{C}$. We aim at finding a *partitioning* of entities $\mathcal{S} = \{S_i\}_{i=1}^{N_c}$, such that each entity is assigned to a single concept at a time, i.e., $\forall S_i, S_j \in \mathcal{S} \rightarrow S_i \cap S_j = \emptyset$, and $\bigcup_{S \in \mathcal{S}} S = \mathcal{E}$. To this end, a natural solution is to perform a *hard clustering* of the entities in the KG $\mathcal{G}$. Then, we can reify the cluster membership relations, i.e., introducing the concepts $c_1, \ldots, c_{N_c}$ as *new entities*, and materializing the concept membership relations as *new triples*. We refer to this process as *KG augmentation*. Algorithm 1 summarizes our framework, which can be instantiated for different clustering and neural link prediction models. We name it Concept Formation via Augmentation (ConFormA). Next, we discuss *how* to perform these two steps, and the reasons *why* neural link predictors can benefit from being trained on augmented KGs.

**Clustering entities.** Ideally, the clustering step in ConFormA could be performed by any relational clustering algorithm. However, classical probabilistic approaches such as statistical predicate invention [Kok and Domingos, 2007] and stochastic block models [Kemp et al., 2006, Xu et al., 2006] would hardly scale to modern KGs with hundreds of thousands of entities. This poses a challenge also to kernel-based approaches [Blondel et al., 2008, de Vries, 2013, Morris et al., 2017]. To overcome this issue, we opt for a more computationally efficient alternative: we first *propositionalize* entities into $d$-dimensional embedding vectors [Kramer et al., 2001] and then employ a propositional clustering algorithm – e.g., spectral clustering [Ng et al., 2001] or K-means – over this now tabular representation $\mathbf{P} \in \mathbb{R}^{N_e \times d}$.

We find that executing multi-hop random paths in $\mathcal{G}$, as proposed by Das et al. [2020], provides scalable and accurate entity representations. We also explored clustering directly over the embeddings learned by a neural link prediction model but with scarce or no link prediction improvements (see Appendix E, Table 15). This could be due to the latent concept information which we explicitly introduce via augmentation being already captured by the neural link predictor embeddings, while graph-based features add complementary information, as observed in Nickel et al. [2016].

**Knowledge Graph Augmentation.** Given the set $\mathcal{S}$, we reify the cluster membership relations by materializing new triples to augment $\mathcal{G}$. Specifically, for each entity $e$ participating in a cluster $c \in \mathcal{C}$ we create a new triple of the form $\langle e, \text{ISA}, c \rangle$, where $c$ is a new entity denoting the $j$-th concept, and ISA is a freshly introduced relation denoting concept memberships.[1]

Let $\mathcal{G}'$ denote the augmented KG, i.e., $\mathcal{G}' \leftarrow \mathcal{G} \cup \{\langle e, \text{ISA}, c \rangle \mid e \in S_c, c \in \mathcal{C}\}$, where $S_c$ is the set of entities assigned to concept $c$. Learning a neural link predictor simply requires calling its usual training routine (lines 6-8 of Algorithm 1) over the augmented KG $\mathcal{G}'$. Specifically, its set of parameters $\mathbf{\Theta}' = \{\mathbf{E}, \mathbf{W}, \mathbf{C}\}$, which now includes the concept embeddings $\mathbf{C} \in \mathbb{C}^{N_c \times k}$ for the newly introduced concept entities $\mathcal{C} = \{c_1, \ldots, c_{N_c}\}$, can be updated by minimizing the neural link predictor's loss function, along its other parameters.

ConFormA is likely to improve the generalization of a neural link predictor for the following reasons. Firstly, this kind of augmentation acts as injecting background knowledge that does not need to be learned from scratch, akin to when inverse relation triples [Lacroix et al., 2018, Kazemi and Poole, 2018] or hierarchical relation information [Zhang et al., 2018] are explicitly added to KGs. Secondly, they help to make very sparse KGs more dense, tackling the sparsity issues in neural link predictors [Pujara et al., 2017].

## 4. ConFormAE: Jointly learning Concepts and Embeddings

ConFormA is a flexible framework: it can be customized with any propositionalization and clustering routines, and wrapped around any neural link prediction model. A natural question then arises: *is it possible to automatically devise a propositionalization scheme that enhances clustering and embedding quality, that is, to jointly learn both the concepts and the embeddings?* Ideally, we could cast this as a joint optimization problem to maximize the marginal log-likelihood of the triples in $\mathcal{G}$, where marginalization is performed over some latent variable $Z$ denoting the cluster assignments, i.e., having values $c \in \mathcal{C}$. As directly maximizing this marginal likelihood is intractable, we adopt an iterative Expectation-Maximization

---

1. A similar augmentation strategy has been independently proposed in Gad-Elrab et al. [2020] to learn rule-based explanations for a subset of the KG entities. See Section 5 for a discussion.

---

**Algorithm 2** ConFormAE($\mathbf{\Theta}, \mathcal{S}, N_c, n, t$)

---

1: **Input:** no. of clusters $N_c$, initial parameters $\mathbf{\Theta}$, no. of epochs $n$, and no. of iterations $t$.
2: **Output:** Updated parameters $\mathbf{\Theta}$, cluster memberships $\mathcal{S} = \{S_c \mid c \in \mathcal{C}\}$
3: **for** $t$ iterations **do**
4:     **for** $c \in \mathcal{C}$ **do**                              $\triangleright$ Initialise the cluster memberships
5:         $S_c \leftarrow \emptyset$
6:     **for** $e \in \mathcal{E}$ **do**
7:         $\hat{c} \leftarrow \arg\max_{c \in \mathcal{C}} \phi_{\text{ISA}}(\mathbf{e}_e, \mathbf{e}_c)$
8:         $S_{\hat{c}} \leftarrow S_{\hat{c}} \cup \{e\}$
9:     $\mathcal{G}' \leftarrow \mathcal{G} \cup \{\langle e, \text{ISA}, c\rangle \mid e \in S_c, c \in \mathcal{C}\}$              $\triangleright$ E-step: Make hard assignments
10:     **for** $n$ epochs **do**              $\triangleright$ M-step: Refine the neural link prediction model
11:         $\mathbf{\Theta} \leftarrow \text{train}(\mathcal{G}', \mathbf{\Theta})$
     **return** $\mathbf{\Theta}, \mathcal{S}$

---

(EM)-like scheme [Dempster et al., 1977]. Algorithm 2 summarizes the whole process, which we name ConFormAE – Concept Formation with Augmentation via EM. We next discuss in detail how to design the expectation (E) and maximization (M) steps efficiently.

**E-step.** Let $p(Z \mid e)$ denote the distribution over concept-memberships induced by the neural link predictor for the entity $e$. Recall from Section 2 that the probability of assigning entity $e$ to concept $c$ is proportional to the score assigned to the reified triple (Section 3) encoding its concept membership, i.e., $p(Z = c \mid e) \propto \phi_{\text{ISA}}(\mathbf{e}_e, \mathbf{e}_c)$, where $\mathbf{e}_e, \mathbf{e}_c$ denote the embeddings of $e$ and $c$, respectively. Exactly computing all the cluster memberships $p(Z = c \mid e)$ for each entity $e$ and concept $c \in \mathcal{C}$ is a hard problem, since we would need to compute an intractable partition function. We therefore resort to compute *hard cluster assignments*, a practical approximation commonly adopted in many *hard-EM variants* [Samdani et al., 2012, Kok and Domingos, 2007]. That is, we are interested in solving $\hat{c} = \arg\max_{c \in \mathcal{C}} p(Z = c \mid e_i)$ for each entity $e_i$. Note that this can be done exactly and efficiently as $\hat{c} = \arg\max_{c \in \mathcal{C}} \phi_{\text{ISA}}(\mathbf{e}_e, \mathbf{e}_c)$ and therefore it reduces to predicting the most probable link between the entity $e$ and a concept in $\mathcal{C}$.

**M-step.** The aim of this step is to find the best set of parameters $\mathbf{\Theta}'$ for a neural link predictor by maximizing its log-likelihood, i.e., $\mathbb{E}_{c_e \sim p(\cdot|e)} \left[\log p(X) + \sum_{e \in \mathcal{E}} \log p(Z = c_e \mid e)\right]$ where $\log p(X)$ here compactly denotes the likelihood of the data in $\mathcal{G}$, and $c_e \in \mathcal{C}$ denotes the concept associated with the entity $e \in \mathcal{E}$. This quantity can be efficiently approximated via our reification and augmentation scheme. In fact, at the end of the E-step, we had retrieved the clustering $\mathcal{S}$ (as in ConForma, cf. Section 3). Therefore, to find $\mathbf{\Theta}'$ we can simply train the neural link prediction model for a certain number of epochs $n$ over the augmented KG $\mathcal{G}'$.

We refer to Appendix A for an in-depth analysis of the time and space complexity of ConForma and ConFormAE.

## 5. Related Work

Concepts are a fundamental building block of modern Knowledge Graphs. For example, the Resource Description Framework [RDF, Klyne and Carroll, 2004] data model allows to state

that a resource is an instance of a concept (or class) via the RDF:TYPE predicate, while its extension RDF Schema [Brickley and Guha, 2014] allows for specifying subclasses between concepts, and using concepts for specifying domain and range of predicates.

**Concept Learning for Relational Data.** Relational clustering, sometimes known as statistical predicate invention [Kok and Domingos, 2007], has been addressed in different communities and under different relational formalisms. For instance, the Infinite Relational Model [IRM, Kemp et al., 2006], is a Bayesian non-parametric interaction method for detecting community of users in networks, later extended to multiple relation types [Xu et al., 2006]. The Multiple Relational Clusterings [MRC, Richardson and Domingos, 2006] model extends the IRM to learning multiple cross-cutting clusterings, i.e. allowing each object to belong to more than one cluster, under the general framework of Markov logic networks (MLNs). The above approaches (and their variants) would hardly scale to the KGs we employ in our experiments. To see why, consider that to train either the IRM or the MRC on a small knowledge graph such as UMLS [McCray, 2003] (135 entities, see Section 6) took 10 hours, while both our algorithms require less than 3 minutes.

**Concept learning for KGE models.** In a preliminary study, Nickel and Tresp [2011] explore how to reconstruct a taxonomy over entities by performing hierarchical clustering of entity embeddings. Differently from our work, they do not learn embedding representations for the learned concepts nor use concept learning to improve link prediction on the original KG. Zhang et al. [2018], on the other hand, proposed learning and explicitly modeling taxonomies over relations. To do so, they cluster relation embeddings in a three-layer hierarchy. Our augmentation approach, while focusing on entity concepts, can be generalized to relations as well, after reifying them. Closer to our work, Gad-Elrab et al. [2020] propose an iterative clustering scheme which involves i) clustering latent embeddings, ii) reifying learned cluster memberships and then, iii) updating latent embeddings. Differently from our work, however, they do not aim at improving neural link prediction accuracy, but eliciting rules to explain subsets of entities. Without this difference in mind, their scheme can be thought of as a particular instance of ConFormA where K-Means is used for directly clustering the latent embeddings. We confirmed empirically that this instance of ConFormA does not yield statistically significant improvements over baselines (see Appendix E, Table 15). More interestingly, Gad-Elrab et al. [2020] propose different ways to reify concept memberships – adopting them in ConFormA is a promising research direction.

**Concept Learning for Deep Graph Classification.** Ying et al. [2018] introduce a differentiable graph pooling module in graph neural networks (GNNs) to for perform hierarchical clustering of nodes in graphs. While improving the accuracy over small-scale graph classification benchmarks, their work cannot be readily adapted to link prediction on KGs and would not scale to large benchmark KGs, such as those used in our experiments.

**Data Augmentation for Link Prediction.** Various KG augmentation schemes have been proposed in the neural link prediction literature. E.g., Lacroix et al. [2018] show that introducing reciprocal relations as explicit triples greatly enhances the performance of KGE models on many benchmarks. Minervini et al. [2017], instead, temporarily augment a KG during training by generating sets of adversarial examples that maximize an inconsistency loss encoding certain background knowledge.

Table 1: Statistics of knowledge graphs. Train, val and test denote the number of facts in the training, validation and test sets, respectively. $|\mathcal{E}|$ and $|\mathcal{R}|$ represent the number of unique entities and relations in the KG. RD and ED are measures of relation and entity density, respectively.

|          | Train   | Valid  | Test   | $|\mathcal{E}|$ | $|\mathcal{R}|$ | RD    | ED |
|----------|---------|--------|--------|--------|--------|-------|----|
| UMLS     | 5,216   | 652    | 661    | 135    | 46     | 113   | 77 |
| WN18RR   | 86,835  | 3,034  | 3,134  | 40,943 | 11     | 7,891 | 4  |
| FB15k-237 | 272,115 | 17,535 | 20,466 | 27,395 | 237    | 1,148 | 38 |

## 6. Experiments

In this Section, we aim to answer the following research questions: **Q1**) are the concepts learned by ConFormA and ConFormAE semantically-meaningful?, **Q2**) can unsupervised concept learning boost neural link prediction performance?, **Q3**) can concept reification help alleviate the cold-start problem in sparse in KGs?, and **Q4**) how does augmentation impact generalization over rare relation types?. We proceed by outlining our experimental setting.

**Datasets.** We perform experiments on three datasets: WN18RR [Dettmers et al., 2018] and FB15k-237 [Toutanova and Chen, 2015] – two large benchmark KGs and UMLS [McCray, 2003] – a small biomedical KG. Summary statistics of the entity and relation distributions for each dataset are shown in Table 1. The two large KGs come with unique challenges – in FB15k-237, the number of predicates is relatively high (Table 1), and it may be difficult to jointly model all of them. In WN18RR, on the other hand, most entities are sparsely represented in the training set. As in [Pujara et al., 2017], we consider the sparsity of each graph by computing the *entity density* (ED) and *relation density* (RD), i.e., the average number of triples per entity or relation: $\text{RD} = |\mathcal{T}|/|\mathcal{R}|$, $\text{ED} = 2|\mathcal{T}|/|\mathcal{E}|$ where $|\mathcal{T}|$ is the number of train triples. We note that the entity density in WN18RR is extremely low, with each entity occurring on average in only four triples (Table 1).

**Baselines.** To investigate the ability of ConFormA and ConFormAE to work with different out-of-the-box neural link predictors, we employ two different baselines: ComplEx [Trouillon et al., 2016] and DistMult [Yang et al., 2015a]. For all experiments we used the nuclear N3 norm [Lacroix et al., 2018] as a regularizer, the standard multi-class loss proposed by Lacroix et al. [2018], and the AdaGrad optimizer [Duchi et al., 2011]. Hyperparameter values can be found in Appendix G. We trained each model till convergence for 100 epochs and computed the filtered Mean Reciprocal Rank (MRR) and HITS@K [Bordes et al., 2013] every 3 epochs on the validation and test sets. The highest validation performance was extracted and the corresponding test performance was reported.

**ConFormA.** We use the propositionalization scheme leveraging random paths proposed by Das et al. [2020]. To construct the vector embeddings, we represent each entity $e \in \mathcal{E}$ using $\mathbf{p}$ where each entry $\mathbf{p}_i$ is given by the number of times we have traveled along relation $r_i$ across the $n$ paths. We distinguish as to whether we have traveled along a relation in the forward or inverse direction, hence the resulting embeddings are $\mathbf{p} \in \mathbb{R}^{2N_r}$. We clustered the resulting representations using the Spectral Clustering algorithm [Ng et al., 2001] using the default parameters, and the number of clusters in $\{50, 100, 500, 1000\}$ for WN18RR and

Table 2: Fragments of prototypical concepts learned for FB15K-237 by ConFormA and for WN18RR by ConFormAE, using ComplEx with $k = 2000$.

| FB15K-237: ConFormA | | | WN18RR: ConFormAE | | |
|---|---|---|---|---|---|
| **Concept 1** | **Concept 2** | **Concept 3** | **Concept 1** | **Concept 2** | **Concept 3** |
| political satire | hypothyroidism | Royal College of Music | mathematics | bird family | russia |
| absurdism | Crohn's disease | Royal Academy of Music | physics | arthropod genus | norway |
| experimental film | yellow fever | Moscow Conservatory | psychology | asterid dicot genus | israel |
| Surrealism | angina pectoris | Manhattan School of Music | computer science | arthropod family | mexico |
| independent film | pancreatitis | Milan Conservatory | chemistry | fish family | antarctica |

FB15k-237, and $\{30, 50, 100\}$ for UMLS. The hyperparameters for training the neural link predictors were selected with the baseline model on a held-out validation set.

**ConFormAE.** To train ConFormAE we experimented with initializing cluster memberships in two ways: randomly and using memberships learned via clustering of simple propositionalized embeddings, such as those used for ConFormA. We found that in most cases a random initialization performed competitively and reduced compute time hence this is the strategy we have opted to use throughout this work. To obtain the ConFormAE results quoted in Table 3 the initial number of clusters was in $\{50, 100, 500, 1000\}$. Our experiments showed that setting $n = 1$ i.e., training the neural predictor for one epoch after every E-step was sufficient for fast convergence (see Appendix G). Again, we use the same hyperparameters used for the baselines. [2]

**Experimental Results.** In order to answer **Q1** we first perform a qualitative analysis: we inspect the entities which form the concepts learned by ConFormA and ConFormAE on UMLS, FB15k-237 and WN18RR. Excerpts of prototypical clusters learned by ConFormA and ConFormAE are shown in Table 2, while full clustering of UMLS and further examples can be found in Appendix C. Across all datasets entities appear to be meaningfully clustered into e.g., diseases, music schools and geographical locations. Next, we strengthen our analysis with a quantitative evaluation: we compare against ground truth concept information related to the semantic types in UMLS [Bodenreider and McCray, 2003] and the `notable_types` in FB15k-237. We report our findings in Appendix D by evaluating cluster matches using normalized mutual information scores, generalized to deal with overlapping clusters for FB15k-237. In summary, for UMLS, we find that ConFormA with random paths and ConFormAE outperform clustering neural link predictor embeddings and recover the group information rather faithfully, with ConFormAE achieving the best scores overall (Table 10). For FB15k-237, we find that across all 3 approaches the NMI is relatively low, indicating that all the methods struggle to recover the concepts as specified by types. Nevertheless, even in this scenario ConFormAE delivers the best scores (Tables 11 and 12).

Hence, we can answer **Q1** affirmatively and note one advantage of ConFormAE over ConFormA– ConFormAE can yield meaningful concepts from random cluster initialization, without requiring any a priori knowledge on the structure of the KG.

To answer **Q2** Table 3 reports the MRR and Hits@$K$ for ConFormA and ConFormAE, for different values of embedding size $k$, after a grid search on regularizers, batch-size and

---

2. Code is available at: https://github.com/AgaDob/conformae

Table 3: MRR and Hits at $K$ (H@$K$) for ConFormA and ConFormAE when using DistMult or ComplEx as baselines on WN18RR and FB15k-237 for different values of embedding size ($k$). Each configuration was repeated with 30 random seeds, and we report the means of each metric. For assessing whether the MRR values are significantly higher than the baseline, we used a one-sided Wilcoxon signed-rank test, where ▲ (resp. △) denotes a $p$-value $\leq 0.01$ (resp. 0.1).

| | $k$ | Model | ComplEx | | | | DistMult | | | |
| | | | MRR | H@1 | H@3 | H@10 | MRR | H@1 | H@3 | H@10 |
|---|---|---|---|---|---|---|---|---|---|---|
| **WN18rr** | 500 | Baseline | 48.06 | 43.71 | 49.73 | 56.57 | 44.23 | 40.21 | 45.34 | 52.72 |
| | | ConFormA | 48.48 ▲ | 43.83 | **50.23** | **57.65** | 44.33 △ | 40.15 | **45.36** | **53.19** |
| | | ConFormAE | **48.51** ▲ | **44.05** | 50.13 | 57.40 | **44.36** ▲ | **40.29** | 45.28 | 53.10 |
| | 1000 | Baseline | 48.58 | 44.18 | 50.21 | 57.16 | 44.53 | 40.34 | 45.68 | 53.20 |
| | | ConFormA | **49.13** ▲ | 44.43 | **50.82** | **58.57** | **45.59** ▲ | **41.25** | **46.76** | **54.72** |
| | | ConFormAE | 48.99 ▲ | **44.45** | 50.60 | 58.12 | 45.15 ▲ | 40.83 | 46.35 | 54.31 |
| | 2000 | Baseline | 48.81 | 44.39 | 50.41 | 57.45 | **45.17** | **40.89** | **46.49** | 53.91 |
| | | ConFormA | **49.28** ▲ | **44.63** | **50.94** | **58.73** | 44.92 | 40.59 | 46.08 | 53.76 |
| | | ConFormAE | 49.14 ▲ | **44.63** | 50.75 | 58.22 | 44.96 | 40.65 | 46.04 | **53.92** |
| **FB15k237** | 500 | Baseline | 35.99 | 26.60 | **39.54** | 54.90 | 34.82 | 25.52 | 38.24 | 53.64 |
| | | ConFormA | 35.97 | 26.55 | 39.50 | **54.98** | 34.86 ▲ | 25.55 | 38.29 | **53.68** |
| | | ConFormAE | **36.06** ▲ | **26.66** | 39.53 | 55.08 | **34.95** ▲ | **25.67** | **38.29** | **53.68** |
| | 1000 | Baseline | 36.11 | 26.68 | 39.65 | 55.15 | 34.95 | 25.61 | 38.42 | 53.73 |
| | | ConFormA | 36.20 △ | 26.69 | 39.71 | 55.20 | 35.32 ▲ | 25.59 | 38.50 | **53.89** |
| | | ConFormAE | **36.25** ▲ | **26.72** | **39.72** | **55.28** | **35.35** ▲ | **25.62** | **38.52** | 53.88 |
| | 2000 | Baseline | 36.26 | 26.83 | 39.79 | 55.33 | 35.39 | 25.99 | 38.86 | 54.37 |
| | | ConFormA | 36.31 ▲ | 26.86 | **39.86** | 55.39 | 35.49 ▲ | 26.11 | 38.89 | **54.49** |
| | | ConFormAE | **36.35** ▲ | **26.95** | 39.84 | **55.44** | **35.50** ▲ | **26.12** | **38.94** | 54.48 |

learning rates for the baselines. We report additional results in Appendix E where we quantitatively inspect which triples benefit most from concept learning and report results with TuckER [Balazevic et al., 2019] – an additional neural link predictor. In general, we see a consistent boost over both DistMult and ComplEx baselines. The boost is especially striking on WN18RR. For example, a smaller ($k = 500$) model learned by ConFormA or ConFormAE is equally good or better than a much larger one ($k = 2000$) learned by ComplEx in terms of in terms of H@10. If we perform additional augmentations, such as adding reciprocal relationships [Lacroix et al., 2018] we find that the improvements from the two methods are additive, as reported in Appendix B. For FB15k-237 we also see an improvement in Table 3, though of a smaller magnitude. This can be explained by the fact that WN18RR is a much sparser KG and as such it can benefit more from our concept learning scheme. Therefore, we hypothesize that the concept reification and explicit augmentation might be especially beneficial for performing link prediction in a small data regime and thus alleviate the *cold-start problem* [Bobadilla et al., 2012]. To verify this (**Q3**), we performed a series of *sparsification* experiments, decreasing the percentage of training triples available to the propositionalisation algorithm and to the link predictor. Fig. 2 and Table 4 show the relative improvement upon the baseline in terms of MRR for the percentage of retained triples in $\{5, 7, 10, 20, \ldots, 90\}$. Across all configurations we see a clear boost, more evident for few training triples. For WN18RR, ConFormA and ConFormAE improve over 80%

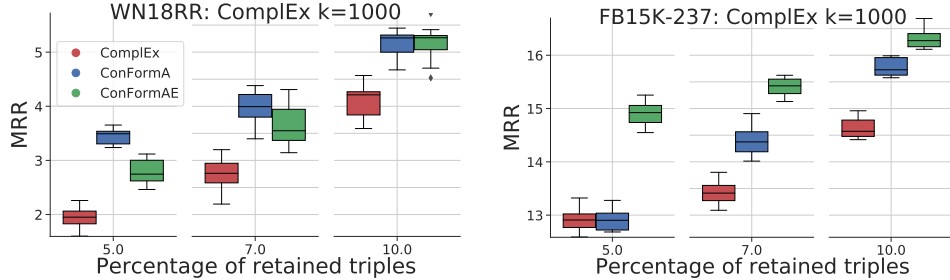

Figure 2: MRR of CONFORMA, CONFORMAE, and ComplEx on sparsified WN18RR and FB15K-237 KGs, where the percentage of training triples is in $\{5, 7, 10\}$ for different values of embedding size ($k$). Each experiment was repeated with 6 different random seeds. Results for other baselines and embedding sizes are shown in Table 25.

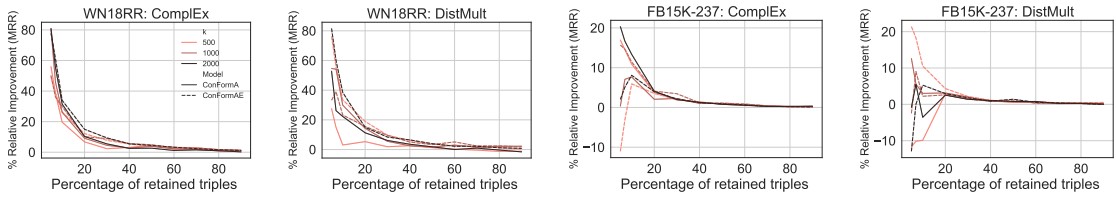

Table 4: Relative improvement in MRR for CONFORMA and CONFORMAE upon the baseline models – either DistMult or ComplEx – on sparsified WN18RR and FB15k-237 KGs for different values of embedding size ($k$). Each experiment was repeated with 6 different random seeds.

w.r.t. their baselines when only 5% of training data is available. For FB15k-237, we also see a consistent boost, though there is some stochasticity for less than 20% training triples - this is likely due to its large number of predicates which increase the minimum number of training triples required to learn a good model.

To answer **Q4**, we inspect how generalization affects different triples after binning them w.r.t. the frequency (rare, medium, common) of their relations. The bins used to categorize relations into their frequency-based sub-populations (Table 26) were constructed by considering the total number of training examples. Fig. 3 reports the relative improvement of CONFORMA and CONFORMAE over the baselines on predicate sub-populations in terms of MRR w.r.t. the baseline for the aforementioned bins. Across all datasets, the largest improvement is observed on the rare predicates sub-population, with relative improvements of up to 8% on WN18RR, 15% on UMLS and 2% on FB15k-237 confirming that discovering concepts and augmenting KGs with them helps neural link predictors on triples whose predicates are underrepresented.

We note that while generally CONFORMA and CONFORMAE perform similarly, occasionally randomly-initialized CONFORMAE performs worse, as can be seen for FB15k-237 in Fig. 3. As is the case for other Hard-EM algorithms, the quality of the clustering learned

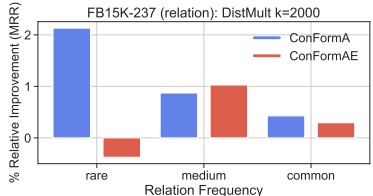
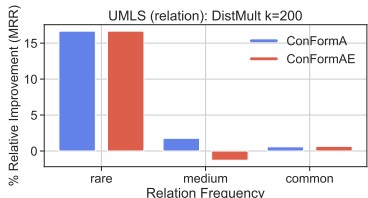

Figure 3: Relative improvement on predicate sub-populations in terms of MRR achieved by ConFormA and ConFormAE over the DistMult baseline on WN18RR and FB15k-237 for $k = 2000$, and on UMLS for $k = 200$ using bins from Table 26.

by ConFormAE depends to a certain extent on the cluster initialization. We find that the cluster memberships learned by ConFormA through clustering of propositionalized representations can provide a good initialization for ConFormAE and accelerate convergence. Strictly speaking, ConFormA can be seen as a special case of ConFormAE with a non-random initialization, where only a series of M-steps is performed.

Lastly, we consider the run-times. Averaging across all embedding sizes, datasets, and six random trials we find that ConFormA and ConFormAE only add a minimal overhead over the respective baselines: they are only 1.09 and 1.14 times slower. For example, training DistMult on WN18RR using $k = 2000$ for 100 epochs took 176min for the baseline, 211min for ConFormA and 227min for ConFormAE with random initialization.

## 7. Conclusions

In this work we have introduced the task of *unsupervised concept formation in KGs* and proposed two algorithms to achieve it – ConFormA and ConFormAE – based on entity clustering and KG augmentation. Our experiments show that our approaches can learn semantically-meaningful concepts and improve the accuracy on downstream link prediction tasks. We find that leveraging latent concept information helps neural link predictors to generalize to rare predicates and is especially beneficial in sparse KGs, where entities participate in few training triples. Moreover, learning new concepts as entities can help to automate the construction of KGs and the learned concept representations can be used for a variety of downstream tasks. While the assumption that every entity participates in exactly one concept can be unrealistic, it points to exciting future work on learning concept hierarchies. Lastly, our work also paves the way for principled probabilistic approaches to elicit discrete latent variables in neural link prediction models.

## Acknowledgments

This research was supported by the European Union's Horizon 2020 research and innovation programme under grant agreement no. 875160.

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

## Appendix A. Computational Complexity

The time and space complexity of ConFormA depends on the propositionalization, clustering, and neural link prediction model being used. In our experiments, the random path propositionalization has time complexity $\mathcal{O}(kL)$, where $L$ is the max length of a path and $k$ is the number of paths, while its space complexity is $\Theta(kN_r)$.

**Clustering.** For a vanilla spectral clustering implementation the complexity would be dominated by the $\mathcal{O}(N_e^3)$ cost of computing the Singular Value Decomposition of the propositionalization embedding matrix $\mathbf{P}$. Alternatively, the K-means algorithm would require $\mathcal{O}(tkN_eN_c)$ time, where $t$ is the number of iterations and $k$ is the embedding size.

**Neural Link Prediction.** For the cost of training and evaluating neural link prediction models, we refer the reader to their respective papers [Bordes et al., 2013, Trouillon et al., 2016, Yang et al., 2015a, Dettmers et al., 2018, Lacroix et al., 2018]. We refer to [Ruffinelli et al., 2020] for a comparison of different choices of the loss function on several downstream link prediction tasks. We point out that, in our case, the number of entities in $\mathcal{G}$ becomes $N_e + N_c$, as the new set of entities in the augmented KG $\mathcal{G}'$ would include $N_c$ concept entities.

**Expectation-Maximization.** In ConFormAE, the complexity of the M-step is that of training the neural link predictor. In the E-step we need to evaluate the score and loop through all of the entities and all the concepts, which results in $\mathcal{O}(N_eN_c)$ steps. Note that, in practice, this step can be efficiently parallelized on GPU. In Section 6 we report average run-times, showing that the computational cost is only marginally higher than that of an out-of-the-box link predictor: in our experiments, ConFormA and ConFormAE are respectively only 1.09 and 1.14 times slower than the neural link predictor alone.

## Appendix B. Reciprocal Relations with ConFormA/E

Reciprocal relations [Lacroix et al., 2018] is a popular method of augmenting KGs by introducing an inverse of every relation into the graph. In Table 5 we compare link prediction performance between training neural link predictors on standard KGs, KGs augmented with inverse relations, training with concept augmentations (ConFormA and ConFormAE) and lastly, the effect of combining inverse relations with ConFormA and ConFormAE. Across all datasets and models we find that best performance is achieved either by combining inverse relations with ConFormA and ConFormAE, or by our method alone.

Table 5: Comparison of link prediction results between neural link predictors and CON-FORMA and CONFORMAE trained on standard KGs and on KGs augmented reciprocal relations. All results are averages of runs with 5 different random seeds for the rank of 1000.

| FB15K-237 | | | WN18RR | | |
|---|---|---|---|---|---|
| Model | Relations | MRR | Model | Relations | MRR |
| ComplEx | Standard | 36.11 | ComplEx | Standard | 48.58 |
| ComplEx | Reciprocal | 36.22 | ComplEx | Reciprocal | 48.62 |
| CONFORMA | Standard | 36.20 | CONFORMA | Standard | 49.13 |
| **CONFORMA** | **Reciprocal** | **36.37** | **CONFORMA** | **Reciprocal** | **49.25** |
| CONFORMAE | Standard | 36.25 | CONFORMAE | Standard | 48.99 |
| CONFORMAE | Reciprocal | 36.28 | CONFORMAE | Reciprocal | 48.96 |
| DistMult | Standard | 35.26 | DistMult | Standard | 44.53 |
| DistMult | Reciprocal | 35.28 | DistMult | Reciprocal | 44.20 |
| CONFORMA | Standard | 34.95 | **CONFORMA** | **Standard** | **45.59** |
| CONFORMA | Reciprocal | 35.34 | CONFORMA | Reciprocal | 45.45 |
| **CONFORMAE** | **Standard** | **35.35** | CONFORMAE | Standard | 45.15 |
| **CONFORMAE** | **Reciprocal** | **35.35** | CONFORMAE | Reciprocal | 45.12 |

Table 6: Fragments of prototypical concepts learned for FB15K-237 by CONFORMAE, using ComplEx with $k = 2000$.

| FB15K-237: CONFORMAE | | | |
|---|---|---|---|
| **Concept 1** | **Concept 2** | **Concept 3** | **Concept 4** |
| Turkey | Ridley Scott | traditional pop music | Academy Award for Best Animated Feature |
| Lithuania | Jerry Bruckheimer | electro house | Golden Globe Award for Best Animated Feature Film |
| Kuwait | Sidney Lumet | electric guitar | Grammy Award for Best Music Film |
| Guatemala | Mike Leigh | post-rock | MTV Video Music Award for Best Pop Video |
| Sri Lanka | Peter Weir | street punk | Grammy Award |

## Appendix C. Qualitative Concepts Analysis

In this section we provide further examples of concepts learned by CONFORMA and CONFORMAE. Tables 6 and 7 show fragments of prototypical concepts learned by CONFORMA for WN18RR and by CONFORMAE for FB15K-237, respectively, while Table 8 and Table 9 show the full clustering of CONFORMAE and CONFORMA for UMLS, respectively.

Table 7: Fragments of prototypical concepts learned for WN18RR by ConFormA, using ComplEx with $k = 2000$.

| WN18RR: ConFormA | | | | |
|---|---|---|---|---|
| **Concept 1** | **Concept 2** | **Concept 3** | **Concept 4** | **Concept 5** |
| carboxyl | counterintelligence | country | spanish-american war | quality |
| aconite | cyber-terrorism | national capital | vietnam war | trait |
| wintergreen oil | terrorism | geographical area | operation desert storm | property |
| protropin | military | city | world war | shape |
| uranyl | bioterrorism | island | battle of britain | skill |

Table 8: Concepts learned by ConFormAE for UMLS, initialized with 50 random clusters, using ComplEx with $k=200$

**Concept 1**
cell_or_molecular-_dysfunction, disease_or_syndrome, experimental_model_of_disease, injury_or_poisoning, mental_or_behavioral-_dysfunction, neoplastic_process, pathologic_function.

**Concept 2**
acquired_abnormality, age_group, anatomical_abnormality, congenital_abnormality, family_group, group, patient_or_disabled_group, population_group, professional_or_occupa--tional_group.

**Concept 3**
biologic_function, cell_function, genetic_function, mental_process, molecular_function, organ_or_tissue_function, natural_phenomenon-_or_process, organism_function, physiologic_function.

**Concept 4**
alga, amphibian, animal, archaen, virus, bacterium, bird, fish, fungus, human, invertebrate, mammal, organism, plant, reptile, ricktteis_or_chlamydia, vertebrate.

**Concept 5**
clinical_attribute, organism_attribute.

**Concept 6**
amino_acid_sequence, body_location_or_region, body_system, carbohydrate_sequence, classification, clinical_drug, conceptual_entity, drug_delivery_device, entity, finding, functional_concept, geographic_area, group_attribute, idea_or_concept, intellectual_product, spatial_concept, laboratory_or_test_result, language, manufactored_object, medical_device, molecular_sequence, nucleotide_sequence, regulation_or_law, research_device, sign_or_symptom.

**Concept 7**
environmental_effect_of-_humans, event, phenomenon_or_process, qualitative_concept, quantitative_concept, temporal_concept, human_caused_phenomenon_or_process.

**Concept 8**
activity, behavior, health_care_activity, daily_or_recreational-_activity, diagnostic_procedure, educational_activity, governmental_or-_regulatory_activity, healt_care_related-_organization, individual_behavior, laboratory_procedure, machine_activity, social_behavior, molecular_biology-_research_technique, occupational_activity, organization, professional_society, research_activity, self_help_or_relief-_organization, therapeutic_or-_preventive_procedure.

**Concept 9**
physical_object.

**Concept 10**
aminoacid_peptide_or-_protein, antibiotic, biologically_active_substance, biomedical_or-_dental_material, body_substance, carbohydrate, chemical, chemical_viewed-_functionally, chemical_view-_structurally, eicosanoid, element_ion_or_isotope, enzyme, food, hazardous_or-_poisonous_substance, hormone, immunologic_factor, indicator_reagent_or-_diagnostic_aid, inorganic_chemical, lipid, neuroactive_substance_or-_biogenic_amine, nucleic_acid_nucleoside-_or_nucleotide, organic_chemical, organophosphorus-_compound, pharmacologic_substance, receptor, steroid, substance, vitamin.

Table 9: Concepts learned by ConFormA for UMLS using ComplEx with $k$=200 by clustering random paths representations using Spectral Clustering, with the number of clusters set to 20.

**Concept 1**
clinical_drug, food, indicator_reagent_or-_diagnostic_aid, chemical, organophosphorus-_compound, chemical_viewed-_functionally, biomedical_or-_dental_material, lipid, chemical_viewed-_structurally, amino_acid_peptide-_or_protein, organic_chemical, carbohydrate, nucleic_acid_nucleoside-_or_nucleotide, element_ion-_or_isotope, steroid, eicosanoid, inorganic_chemical

**Concept 2**
conceptual_entity, spatial_concept, activity, idea_or_concept.

**Concept 3**
cell_component, body_location_or_region, body_substance, anatomical_structure, body_space_or_junction, gene_or_genome, fully_formed-_anatomical_structure, tissue, cell, embryonic_structure, body_part_organ-_or_organ_component.

**Concept 4**
regulation_or_law, classification, intellectual_product.

**Concept 5**
molecular_sequence, language, body_system, carbohydrate_sequence, nucleotide_sequence, amino_acid_sequence, functional_concept.

**Concept 6**
plant, alga, bacterium, fungus, rickettsia_or_chlamydia, virus.

**Concept 7**
quantitative_concept, laboratory_or_test_result.

**Concept 8**
qualitative_concept, finding, sign_or_symptom.

**Concept 9**
research_device, drug_delivery_device, manufactured_object, medical_device.

**Concept 10**
molecular_biology-_research_technique, diagnostic_procedure, research_activity, laboratory_procedure.

**Concept 11**
invertebrate, archaeon, organism, bird, fish, amphibian, animal, reptile, human, mammal, vertebrate.

**Concept 12**
occupation_or_discipline, biomedical_occupation-_or_discipline.

**Concept 13**
organism_attribute, natural_phenomenon-_or_process, temporal_concept, mental_process, genetic_function, molecular_function, biologic_function, cell_function, organ_or_tissue_function, physiologic_function, organism_function.

**Concept 14**
population_group, professional_or-_occupational_group, machine_activity, group_attribute, group, age_group, family_group, patient_or_disabled_group.

**Concept 15**
clinical_attribute, anatomical_abnormality, acquired_abnormality, congenital_abnormality, health_care_activity, injury_or_poisoning, pathologic_function, experimental_model-_of_disease, mental_or_behavioral-_dysfunction, disease_or_syndrome, neoplastic_process, cell_or_molecular-_dysfunction, therapeutic_or-_preventive_procedure.

**Concept 16**
event, entity, physical_object, substance.

**Concept 17**
human_caused-_phenomenon_or_process, phenomenon_or_process, environmen-tal_effect_of_humanst.

**Concept 18**
professional_society, organization, health_care-_related_organization, self_help_or-_relief_organization.

**Concept 19**
governmental_or-_regulatory_activity, educational_activity, geographic_area, behavior, daily_or_recreational-_activity, social_behavior, occupational_activity, individual_behavior.

**Concept 20**
hazardous_or-_poisonous_substance, antibiotic, neuroreactive_substance-_or_biogenic_amine, pharmacologic_substance, vitamin, hormone, immunologic_factor, enzyme, receptor, biologi-cally_active_substance.

# Appendix D. Quantitative Cluster Analysis

## D.1 UMLS

To quantify the cluster quality, we compare our learned clusters against ground-truth clusters available for UMLS and FB15k-237. Specifically, for UMLS we utilize the *semantic group* information [Bodenreider and McCray, 2003], which is composed of groups such as Physiology, Living Beings, Concepts & Ideas, and Chemicals & Drugs. Altogether, semantic groups amount to 14 disjoint clusters, constructed on the basis of semantic validity, parsimony, completeness and utility. Table 10 summarizes our results: CONFORMA and CONFORMAE, both with random paths initialization, score higher NMI than clustering directly over neural link predictor embeddings. By manually inspecting the clusters, we can furthermore say that all approaches recover the group information rather faithfully.

## D.2 FB15K-237

To assess the quality of concepts learned for FB15K-237 we utilize the `notable_type` information available in Freebase. Extracting this information for all entities in FB15K-237 gives rise to 3.8k overlapping clusters, with cluster sizes ranging from 14.5k entities to single-entity clusters. Comparing against the entire set of the overlapping ground-truth clusters would not be very informative in our case, as in this work we have restricted ourselves to smaller numbers of disjoint clusters. Instead, we propose the following two approaches:

1. **Comparison against Top 100 Clusters** We select the 100 largest clusters from the ground truth clustering – jointly, these cover all of the entities in FB15K-237, some more than once. The resulting clusters range from concepts such as *abstract*, *animate* and *person*, through to *award nominee*, *film* and *educational institution*. In order to compare the learned clusterings against the ground truth, we compute the Normalized Mutual Information (NMI) for overlapping clusters [Lancichinetti et al., 2009], comparing the quality of clusters obtained through clustering vanilla ComplEx and DistMult embeddings against random-paths-initialized CONFORMA and random-paths-initialized CONFORMAE. The results are reported in Table 11.

2. **Boolean Cluster Matching** While comparing only against the largest clusters can assess how well the clusters capture global properties of the entities, it does not allow us to assess the quality of more fine-grained clusters. To address this, we propose the following comparison: firstly, we describe each cluster using a Boolean vector such that each entry of the vector corresponds to the presence or absence of a given entity in a cluster. Next, we use the Jaccard Index to find for each learned cluster the closest corresponding cluster in the set of ground truth clusters. Lastly, we compute NMI between the entire clustering learned by random-paths-initialized CONFORMA or random-paths-initialized CONFORMAE, and the selected set of ground-truth clusters. The resulting NMIs can be found in Table 12.

We find that across all 3 approaches the NMI is relatively low, indicating that all the methods struggle to recover the ground truth information as defined by the notable type information in FB. It is worth noting, however, that CONFORMA with relatively low-dimensional random paths representations, $\mathbf{p} \in \mathbb{R}^{2N_r}$ where $N_r = 237$ for FB15K-237,

Table 10: Normalized Mutual Information (NMI) between the learned clustering and the semantic group information for UMLS [Bodenreider and McCray, 2003]. Baselines were obtained by clustering the vanilla ComplEx and DistMult embeddings using Spectral Clustering with the number of clusters set to 15. ConFormA clusterings were obtained by clustering random paths representations. *It is worth noting that the performance of random paths* ConFormA *clustering is independent of the neural link predictor used.* ConFormAE clusters were obtained by initializing them with clustered random paths. Across all runs neural link predictor embeddings of rank 200 were used.

| Model | Baseline | ConFormA | ConFormAE |
|---|---|---|---|
| ComplEx | 0.766 | 0.774 | **0.788** |
| DistMult | 0.754 | 0.774 | **0.798** |

Table 11: Normalized Mutual Information (NMI) between the learned clustering and top 100 largest clusters in *notable type* clustering on FB15k-237. Baselines were obtained by clustering the vanilla ComplEx and DistMult embeddings using Spectral Clustering with the number of clusters set to 100. ConFormA clusterings were obtained by clustering random paths representations. *It is worth noting that the performance of random paths* ConFormA *clustering is independent of the neural link predictor used.* ConFormAE clusters were initialized using clustered random paths. Results are shown for model rank, $k$, in {500, 1000, 2000}.

| Model | K | Baseline | ConFormA | ConFormAE |
|---|---|---|---|---|
| | 500 | 0.104 | 0.121 | **0.150** |
| ComplEx | 1000 | 0.094 | 0.121 | **0.151** |
| | 2000 | 0.089 | **0.121** | 0.094 |
| | 500 | 0.110 | 0.121 | **0.166** |
| DistMult | 1000 | 0.106 | 0.121 | **0.129** |
| | 2000 | 0.101 | 0.121 | **0.157** |

consistently outperforms clustering even significantly larger neural link predictor embeddings. Furthermore, across both, Table 11 and Table 12, ConFormAE initialized with random paths representations improves upon nearly all other scores.

Qualitatively inspecting the cluster matches with the highest Jaccard scores (Table 13) we find, as expected, that the model learns a concept which is very similar to the ground truth. In the case of the lowest Jaccard scores (Table 14) we find that despite the scores being low, concepts learned by ConFormA and ConFormAE are not in themselves poor quality or random. Rather, the issue lies in that the closest corresponding `notable_types` cluster encodes a very different meaning, resulting in poor overlap between the entities.

Table 12: Normalized Mutual Information (NMI) between the learned clustering and 100 clusters selected from the *notable type* database via matching clusters using the Jaacard Index. Baselines were obtained by clustering the vanilla ComplEx and DistMult embeddings using Spectral Clustering with the number of clusters set to 100. CONFORMA clusterings were obtained by clustering random paths representations. *It is worth noting that the performance of random paths* CONFORMA *clustering is independent of the neural link predictor used.* CONFORMAE clusters were initialized using clustered random paths.

| MODEL | K | BASELINE | CONFORMA | CONFORMAE |
|---|---|---|---|---|
| | 500 | 0.214 | 0.251 | **0.311** |
| COMPLEX | 1000 | 0.213 | 0.251 | **0.294** |
| | 2000 | 0.181 | **0.251** | 0.242 |
| | 500 | 0.175 | 0.251 | **0.367** |
| DISTMULT | 1000 | 0.187 | 0.251 | **0.276** |
| | 2000 | 0.176 | 0.251 | **0.360** |

## Appendix E. Link Prediction Results

In this section we provide additional link prediction performance results of our method. In Table 15 we report experiments investigating the impact of performing link prediction with CONFORMA using concepts learned via directly clustering ComplEx embeddings. We find that in most cases the augmentation either degrades the performance or there is no significant improvement. Tables 17 to 22 show top 10 test triples for which our method achieves greatest improvement over the baseline neural link predictor. In Table 16 we visualize link prediction results shown earlier in tabular form (Table 3) and in Table 23 we provide link prediction performance for UMLS. In Table 24 we include link prediction results with Tucker, an additional neural link prediction model. Lastly, in Table 25 we provide an extension of results for link prediction on sparsified KGs, showing results for both, ComplEx and DistMult, for a range of embedding sizes.

### E.1 Link Prediction with TuckER

To further demonstrate that our method can improve upon a wide range of neural link predictors and embedding sizes, we report link prediction results with TuckER [Balazevic et al., 2019] – a recent neural link predictor which has achieved competitive performance for small embedding sizes.

To compute the TuckER baselines for ranks $k$ in $\{50, 100, 500\}$, following the training set-up described in [Balazevic et al., 2019], we used the Adam optimizer [Kingma and Ba, 2015] and performed a gridsearch over the learning rates in $\{0.1, 0.05, 0.01, 0.005, 0.001, 0.0005, 0.0001\}$. We set the predicate embedding size equal to the entity embedding size for both datasets. A batch-size of 128 and learning rate decay of 1.0 were held constant for all experiments.

Table 13: Examples of concepts learned by ConFormA and ConFormAE for FB15K-237 alongside their ground-truth examples, with some of the highest Jaccard Index scores in the entire clustering – 1.0 for ConFormAE and 0.952 for ConFormA, using ComplEx with k=500 and 100 clusters.

| Ground Truth | ConFormA | Ground Truth | ConFormAE |
|---|---|---|---|
| <film.film_ festival_event> | Concept_72 | <sports.sports _league_draft> | Concept_36 |
| 2010 Sundance Film Festival, | 2009 Sundance Film Festival, | 2005 Major League Baseball draft, | 2005 Major League Baseball draft, |
| 1982 Cannes Film Festival, | 2000 Cannes Film Festival, | 2005 NFL Draft, | 2005 NFL Draft, |
| 62nd Berlin International Film Festival, | 2009 Toronto International Film Festival, | 2007 NBA Draft, | 2007 NBA Draft, |
| 39th Berlin International Film Festival, | 2008 Toronto International Film Festival, | 2004 NFL Draft, | 2004 NFL Draft, |
| 2011 Sundance Film Festival, | 59th Berlin International Film Festival, | 2006 Major League Baseball draft, | 2006 Major League Baseball draft, |
| 2011 Toronto International Film Festival, | 1982 Cannes Film Festival, | 2002 Major League Baseball draft, | 2002 Major League Baseball draft, |
| 2009 Sundance Film Festival, | 32nd Berlin International Film Festival, | 2003 NFL Draft, | 2003 NFL Draft, |
| 2008 Sundance Film Festival, | 34th Berlin International Film Festival, | 2006 NFL Draft, | 2006 NFL Draft, |
| 2012 Sundance Film Festival, | 39th Berlin International Film Festival, | 2005 NBA Draft, | 2005 NBA Draft, |
| 34th Berlin International Film Festival, | 60th Berlin International Film Festival, | 2003 NBA Draft, | 2003 NBA Draft, |
| 2009 Toronto International Film Festival, | 2010 Toronto International Film Festival, | 2007 NFL Draft, | 2007 NFL Draft, |
| 59th Berlin International Film Festival, | 58th Berlin International Film Festival, | 1997 Major League Baseball draft, | 1997 Major League Baseball draft, |
| 2008 Toronto International Film Festival, | 2010 Sundance Film Festival, | 2004 NBA Draft, | 2004 NBA Draft, |
| 32nd Berlin International Film Festival, | 2011 Sundance Film Festival, | 1995 Major League Baseball draft, | 1995 Major League Baseball draft, |
| 58th Berlin International Film Festival, | 61st Berlin International Film Festival, | 2008 NBA Draft, | 2008 NBA Draft, |
| 2000 Cannes Film Festival, | 2011 Toronto International Film Festival, | 2004 Major League Baseball draft, | 2004 Major League Baseball draft, |
| 60th Berlin International Film Festival, | 2012 Sundance Film Festival, | 2003 Major League Baseball draft, | 2003 Major League Baseball draft, |
| 2012 Toronto International Film Festival, | 62nd Berlin International Film Festival, | 2007 Major League Baseball draft, | 2007 Major League Baseball draft, |
| 61st Berlin International Film Festival, | 2012 Toronto International Film Festival | 2006 NBA Draft, | 2006 NBA Draft, |
| 2010 Toronto International Film Festival | | 2008 NFL Draft | 2008 NFL Draft |

Link prediction results with TuckER are shown in Table 24, where we find that ConFormA and ConFormAE consistently outperform the baseline TuckER model for nearly all of the configurations.

Table 14: Examples of concepts learned by ConFormA and ConFormAE for FB15K-237 alongside their ground-truth examples, with some of the lowest Jaccard Index scores in the entire clustering – 0.111 for ConFormAE and 0.036 for ConFormA, using ComplEx with k=500 and 100 clusters.

| Ground Truth | ConFormAE | Ground Truth | ConFormA |
|---|---|---|---|
| `<default_domain.facts-_from_the_community>` Republican Party | `Concept_84` Communist Party of the Soviet Union, Communist Party of India (Marxist), Kuomintang, Republican Party, Whig Party, Democratic-Republican Party, Federalist Party, Democratic Party, Canadian Alliance. | `<terrorism.terrorist _organization>` al-Qaeda, Hamas, Hezbollah | `Concept_11` Austria-Hungary, Kingdom of Great Britain, Byzantine Empire, Empire of Japan, Kingdom of Naples, Russian Soviet Federative Socialist Republic, Kingdom of Romania, Spanish Empire, Kingdom of Sardinia, Prussia, Kingdom of Portugal, House of Plantagenet, Kingdom of Italy, Hamas... |

## Appendix F. Datasets

For each of the datasets used to evaluate our approach – UMLS, WN18RR and FB15K-237 – we provide the frequency bins shown in Table 26 used to divide relations into frequency sub-populations for computing the link prediction results in Fig. 3.

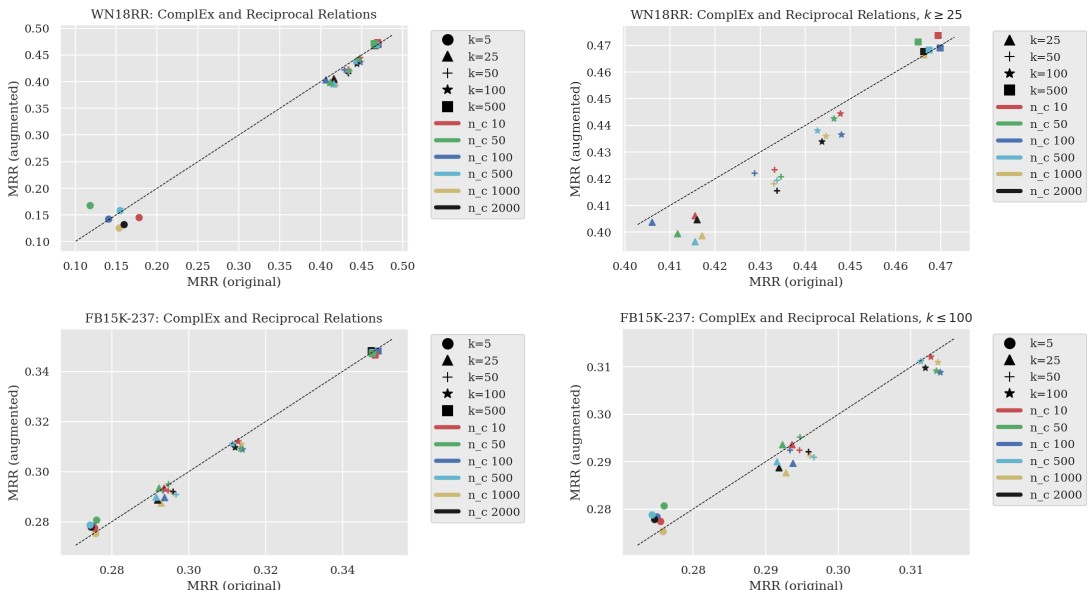

Table 15: Performing ConFormA using concepts learned via clustering of ComplEx embeddings for WN18RR and FB15K-237, for a range of embedding sizes ($k$) and varying the number of clusters ($N_c$), implemented with reciprocal relations. The x-axis corresponds to the performance of the baseline ComplEx model while the y-axis shows the performance of ConFormA. Points appearing above the diagonal line indicate outperforming the baseline. Plots in the right column magnify selected regions of plots in the left column.

## Appendix G. Training Details

### G.1 Baselines

To obtain the best parameters for the baselines, we ran the same grid search for all of the datasets on both, ComplEx and DistMult, with the ranks set to $\{50, 100, 200\}$ for UMLS and $\{500, 1000, 2000\}$ for WN18RR and FB15K-237, using the standard train/validation/test splits. The grid consisted of three batch-sizes in $\{50, 100, 500\}$, three learning rates: $\{10^{-1}, 10^{-2}, 10^{-3}\}$ and six regularization strengths in $\{10^{-3}, 5 \times 10^{-3}, \ldots, 10^{-1}, 5 \times 10^{-1}\}$.

### G.2 ConFormA

**Propositionalisation** To generate the representations we explored the parameter range suggested by Perozzi et al. [2014], using a *minimum path length* of 2, *maximum path length* in $\{3, 5, 10, 20, 30\}$ and two *number of paths* parameters: 32 and 64. We found that the maximum path length parameter was most influential in determining how well a representation captured the characteristics of a given KG. For both, WN18RR and FB15K-237, we found that setting maximum path length to 5 and number of paths to 64 gave competitive results.

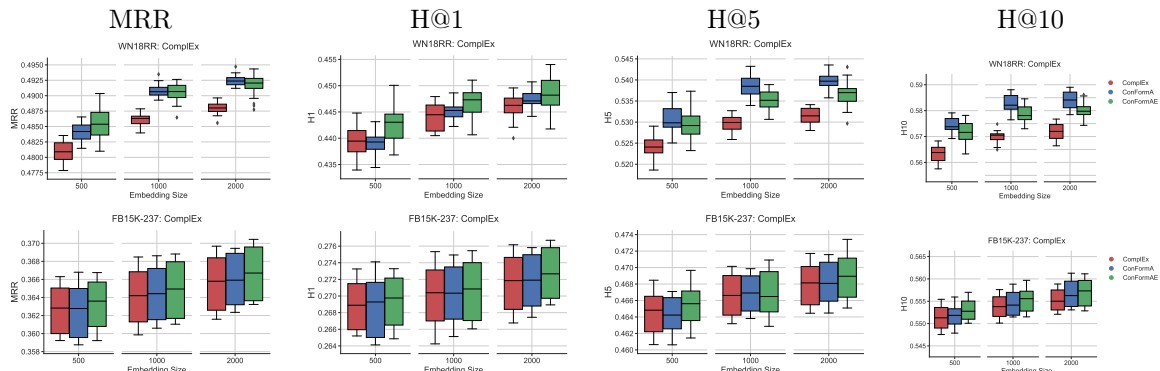

Table 16: MRR and Hits (H) at 1, 5, 10, 50 for ConFormA and ConFormAE when using ComplEx as baseline models on WN18RR and FB15K-237 KGs for different values of embedding size ($k$). Each configuration has been repeated with 30 different random seeds.

Table 17: Top 10 UMLS test triples with greatest improvement of ConFormA over baseline ComplEx model, for $k = 50$.

| UMLS Test Triple | | | Reciprocal Rank | |
|---|---|---|---|---|
| $s$ | $p$ | $o$ | Baseline | ConFormA |
| mental_process | isa | organism_function | 0.06 | 1.0 |
| disease_or_syndrome | occurs_in | mental_or_behavioral_dysfunction | 0.2 | 1.0 |
| finding | isa | conceptual_entity | 0.25 | 1.0 |
| steroid | interacts_with | eicosanoid | 0.33 | 1.0 |
| tissue | adjacent_to | body_space_or_junction | 0.33 | 1.0 |
| embryonic_structure | location_of | virus | 0.33 | 1.0 |
| machine_activity | isa | activity | 0.33 | 1.0 |
| antibiotic | interacts_with | biologically_active_substance | 0.5 | 1.0 |
| congenital_abnormality | complicates | anatomical_abnormality | 0.5 | 1.0 |
| carbohydrate | affects | mental_process | 0.5 | 1.0 |
| laboratory_or_test_result | co-occurs_with | sign_or_symptom | 0.5 | 1.0 |

**Clustering Algorithm** We experimented with clustering the propositionalised representations using a number of clustering algorithms: K-Means, Spectral Clustering, Affinity Propagation and DBSCAN. Across these, we have found no significant difference in performance in terms of both, cluster quality and downstream link prediction performance, hence we used Spectral Clustering for all experiments.

**Number of clusters** The number of clusters was treated as a hyperparameter and chosen from $\{50, 100, 500, 1000\}$ for FB15K-237 and WN188R, and from $\{30, 50, 100\}$ for UMLS. As shown in Fig. 4, lower numbers of clusters on average resulted in better link prediction performance.

Table 18: Top 10 UMLS test triples with greatest improvement of ConFormAE over baseline ComplEx model, for $k = 50$.

| UMLS Test Triple | | | Reciprocal Rank | |
|---|---|---|---|---|
| $s$ | $p$ | $o$ | Baseline | ConFormAE |
| mental_process | isa | organism_function | 0.06 | 1.0 |
| environmental_effect_of_humans | isa | phenomenon_or_process | 0.14 | 1.0 |
| cell | part_of | body_part_organ_or_organ_component | 0.2 | 1.0 |
| neuroreactive_substance_or_biogenic_amine | isa | biologically_active_substance | 0.2 | 1.0 |
| human_caused_phenomenon_or_process | isa | event | 0.25 | 1.0 |
| fully_formed_anatomical_structure | location_of | virus | 0.25 | 1.0 |
| steroid | interacts_with | eicosanoid | 0.33 | 1.0 |
| cell_component | location_of | body_space_or_junction | 0.33 | 1.0 |
| cell_component | location_of | body_space_or_junction | 0.33 | 1.0 |
| organism_function | produces | hormone | 0.33 | 1.0 |
| therapeutic_or_preventive_procedure | complicates | pathologic_function | 0.5 | 1.0 |

Table 19: Top 10 WN18RR test triples with greatest improvement of ConFormA over baseline ComplEx model, for $k = 500$.

| WN18RR Test Triple | | | Reciprocal Rank | |
|---|---|---|---|---|
| $s$ | $p$ | $o$ | Baseline | ConFormA |
| latin.n.03 | _hypernym | person.n.01 | 0.0 | 1.0 |
| periwinkle.n.02 | _hypernym | herb.n.01 | 0.0 | 1.0 |
| threepence.n.01 | _hypernym | coin.n.01 | 0.0 | 1.0 |
| stress.n.03 | _hypernym | emphasis.n.01 | 0.01 | 1.0 |
| trade_name.n.01 | _member_of_domain_usage | clomiphene.n.01 | 0.01 | 1.0 |
| red-winged_blackbird.n.01 | _hypernym | new_world_blackbird.n.01 | 0.01 | 1.0 |
| libel.n.01 | _synset_domain_topic_of | law.n.01 | 0.09 | 1.0 |
| merginae.n.01 | _member_meronym | mergus.n.01 | 0.14 | 1.0 |
| saxifraga.n.01 | _member_meronym | saxifrage.n.01 | 0.14 | 1.0 |
| regimentals.n.01 | _hypernym | military_uniform.n.01 | 0.17 | 1.0 |
| new_zealand.n.01 | _member_of_domain_region | returning_officer.n.01 | 0.2 | 1.0 |

### G.3 ConFormAE

**Number of epochs, $n$, per E-Step:** To explore the effect of changing the number of epochs, $n$, per E-step, we trained a range models and embedding sizes, varying $n$ in $\{1, 2, 3, 5\}$. We found that $n = 1$ provided a good compromise in terms of link prediction performance across different embedding sizes, datasets and neural link predictors - a visualization for WN18RR can be found in Figure Fig. 5.

**Cluster initialization:** Between the two initialization methods we experimented with – random initialization and Spectral Clustering of random paths propositionalisation – we found no significant difference in performance, hence we used random initialization by default due to its reduced complexity, unless the contrary is specified.

**Number of clusters:** We used the same ranges to select the number of clusters as for ConFormA. It is worth noting, however, that while for ConFormA the number of clusters is fixed, in ConFormAE that number of clusters can decrease during training if no entities

Table 20: Top 10 WN18RR test triples with greatest improvement of CONFORMAE over baseline ComplEx model, for $k = 500$.

| WN18RR Test Triple | | | Reciprocal Rank | |
|---|---|---|---|---|
| s | p | o | Baseline | CONFORMAE |
| periwinkle.n.02 | _hypernym | herb.n.01 | 0.0 | 1.0 |
| trade_name.n.01 | _member_of_domain_usage | clomiphene.n.01 | 0.01 | 1.0 |
| ranunculaceae.n.01 | _member_meronym | isopyrum.n.01 | 0.04 | 1.0 |
| shiite.n.01 | _hypernym | muslim.n.01 | 0.05 | 1.0 |
| sapindaceae.n.01 | _member_meronym | genus_harpullia.n.01 | 0.08 | 1.0 |
| libel.n.01 | _synset_domain_topic_of | law.n.01 | 0.09 | 1.0 |
| right_to_vote.n.01 | _synset_domain_topic_of | law.n.01 | 0.12 | 1.0 |
| united_states.n.01 | _has_part | missouri.n.02 | 0.14 | 1.0 |
| compositae.n.01 | _member_meronym | balsamorhiza.n.01 | 0.17 | 1.0 |
| cupressaceae.n.01 | _member_meronym | taxodium.n.01 | 0.17 | 1.0 |
| plural.n.01 | _member_of_domain_usage | sunglasses.n.01 | 0.17 | 1.0 |

Table 21: Top 10 FB15K-237 test triples with greatest improvement of CONFORMA over baseline ComplEx model, for $k = 500$.

| FB15K-237 Test Triple | | | Reciprocal Rank | |
|---|---|---|---|---|
| s | p | o | Baseline | CONFORMA |
| Ocean Software | /business/[...]/industry | video game | 0.0 | 1.0 |
| Alaska | /location/[...]/contains | Nome Census Area | 0.01 | 1.0 |
| Republican Party | /government/[...]/politician | Kevin Smith | 0.03 | 1.0 |
| Bancroft Prize | /award/[...]/category_of | Bancroft Prize | 0.03 | 1.0 |
| Kate Hudson | /people/[...]/type_of_union | domestic partnership | 0.03 | 1.0 |
| Jacqueline Bisset | /people/[...]/gender | female organism | 0.05 | 1.0 |
| Slumdog Millionaire | /film/[...]/film_release_region | United States of America | 0.05 | 1.0 |
| Phil LaMarr | /people/[...]/profession | actor | 0.05 | 1.0 |
| FilmFlex | /film/[...]/film | Night at the Museum | 0.06 | 1.0 |
| The Portrait of a Lady | /film/film/genre | film adaptation | 0.07 | 1.0 |
| Australia | /location/[...]/currency | Australian dollar | 0.07 | 1.0 |

are assigned to some concepts. The number of clusters specified in Fig. 4 for CONFORMAE corresponds to the initial number of clusters.

### G.4 Evaluation

During evaluation, we only consider the triples and entities appearing in the original dataset, to make sure the evaluation metrics for CONFORMA and CONFORMAE are computed using exactly the same protocol as for the baselines.

Table 22: Top 10 FB15K-237 test triples with greatest improvement of CONFORMAE over baseline ComplEx model, for $k = 500$.

| FB15K-237 Test Triple | | | Reciprocal Rank | |
|---|---|---|---|---|
| s | p | o | Baseline | CONFORMAE |
| Ocean Software | /business/[...]/industry | video game | 0.0 | 1.0 |
| Republican Party | /government/[...]/politician | Kevin Smith | 0.03 | 1.0 |
| Bancroft Prize | /award/[...]/category_of | Bancroft Prize | 0.03 | 1.0 |
| The Untouchables | /film/[...]/genre | crime fiction | 0.03 | 1.0 |
| Slumdog Millionaire | /film/[...]e/film_release_region | United States of America | 0.05 | 1.0 |
| Seattle University | /education/[...]/student | Duff McKagan | 0.06 | 1.0 |
| Ryan Reynolds | /people/[...]/nationality | Canada | 0.06 | 1.0 |
| Satellite Awards 2008 | /award/[...]/award_winner | Tom McCarthy | 0.08 | 1.0 |
| Omaha | /location/[...]/time_zones | Central Time Zone | 0.09 | 1.0 |
| Mr. Nobody | /film/[...]/film_release_region | Finland | 0.09 | 1.0 |
| The Best Exotic Marigold Hotel | /film/[...]/film_release_region | Finland | 0.09 | 1.0 |

Table 23: Mean Reciprocal Rank (MRR) and Hits at $K$ (H@$K$) for CONFORMA and CONFORMAE when using DistMult or ComplEx as baselines on UMLS for different values of embedding size ($k$). Each configuration was repeated with 30 random seeds, and we report the means of each metric.

| | $k$ | MODEL | COMPLEX | | | | DISTMULT | | | |
|---|---|---|---|---|---|---|---|---|---|---|
| | | | MRR | H@1 | H@3 | H@10 | MRR | H@1 | H@3 | H@10 |
| UMLS | 50 | BASELINE | 94.64 | 90.92 | **98.26** | 99.59 | 75.50 | 66.62 | 81.01 | 91.70 |
| | | CONFORMA | 94.74 | 91.10 | 98.19 | 99.59 | 75.36 | 66.47 | 80.83 | **91.72** |
| | | CONFORMAE | **95.12** | **91.92** | 98.06 | **99.63** | **75.76** | **67.14** | **81.13** | 91.67 |
| | 100 | BASELINE | 95.45 | 92.34 | 98.44 | **99.66** | **76.13** | **68.28** | 80.65 | 91.42 |
| | | CONFORMA | **95.68** | **92.77** | 98.46 | 99.64 | 76.11 | 68.07 | 80.56 | 91.59 |
| | | CONFORMAE | 95.57 | 92.56 | **98.48** | 99.63 | 76.05 | 67.85 | **80.98** | **91.66** |
| | 200 | BASELINE | 96.46 | 94.08 | **98.84** | 99.72 | 76.21 | 68.37 | 80.69 | 91.36 |
| | | CONFORMA | 96.29 | 93.77 | 98.68 | 99.70 | **76.43** | **68.92** | 80.52 | 91.43 |
| | | CONFORMAE | **96.47** | **94.09** | 98.67 | **99.75** | 76.16 | 68.25 | **80.85** | **91.59** |

Table 24: Link prediction performance of TuckER with CONFORMA and CONFORMAE evaluated on WN18RR and FB15K-237, for a range of embedding sizes, $k$ in $\{50, 100, 500\}$. Each experiment was repeated with 6 different random seeds.

| DATASET | $k$ | MODEL | MRR | H@1 | H@3 | H@5 | H@10 | H@50 |
|---|---|---|---|---|---|---|---|---|
| FB15K-237 | 50 | TUCKER | 28.96 | 20.59 | 31.64 | 37.62 | 45.86 | 64.43 |
| | | CONFORMA | **29.06** | **20.71** | **31.73** | **37.65** | **45.88** | **64.52** |
| | | CONFORMAE | 29.01 | 20.66 | **31.73** | 37.55 | 45.77 | 64.33 |
| | 100 | TUCKER | 30.14 | **21.74** | 32.87 | **38.89** | **47.23** | 65.52 |
| | | CONFORMA | **30.15** | 21.62 | 32.76 | 38.78 | 47.01 | 65.22 |
| | | CONFORMAE | 30.05 | 21.57 | **32.91** | 38.86 | 47.21 | **65.58** |
| | 500 | TUCKER | 32.85 | 24.11 | 36.08 | 42.17 | 50.40 | 67.74 |
| | | CONFORMA | 32.98 | 24.24 | 36.19 | 42.35 | 50.46 | **68.09** |
| | | CONFORMAE | **33.05** | **24.28** | **36.34** | **42.42** | **50.54** | 67.93 |
| WN18RR | 50 | TUCKER | 43.59 | **40.86** | 44.73 | 46.22 | 48.54 | 54.08 |
| | | CONFORMA | **43.65** | 40.81 | **44.74** | **46.56** | **48.93** | **54.83** |
| | | CONFORMAE | 43.18 | 40.41 | 44.46 | 46.00 | 48.10 | 53.73 |
| | 100 | TUCKER | 45.10 | 42.27 | 46.31 | 48.10 | 50.33 | 55.32 |
| | | CONFORMA | **45.44** | **42.39** | **46.80** | **48.69** | **51.07** | **56.68** |
| | | CONFORMAE | 44.84 | 42.02 | 46.08 | 47.72 | 50.04 | 55.20 |
| | 500 | TUCKER | 46.01 | 42.29 | 48.00 | 50.05 | 52.53 | 58.35 |
| | | CONFORMA | **46.59** | **42.72** | **48.64** | **50.71** | **53.51** | **59.63** |
| | | CONFORMAE | 46.08 | 42.37 | 48.09 | 50.02 | 52.60 | 58.28 |

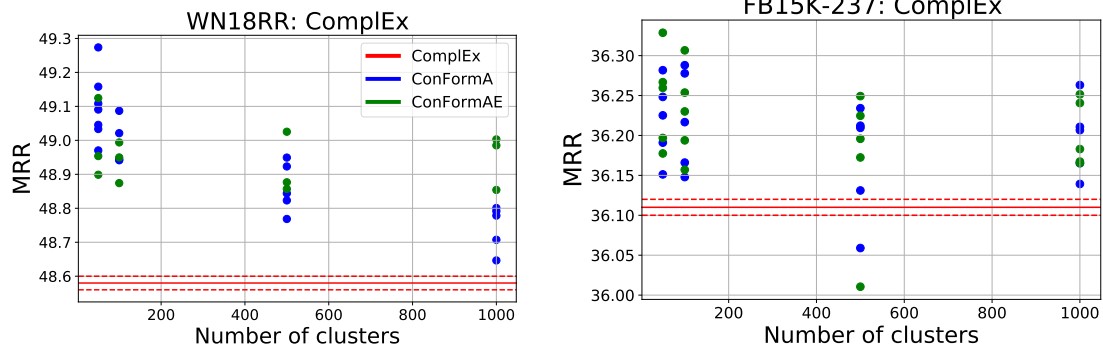

Figure 4: The effect of varying the number of concepts on Mean Reciprocal Rank (MRR), shown for WN18RR and FB15K-237 with ComplEx, using rank size of 1000.

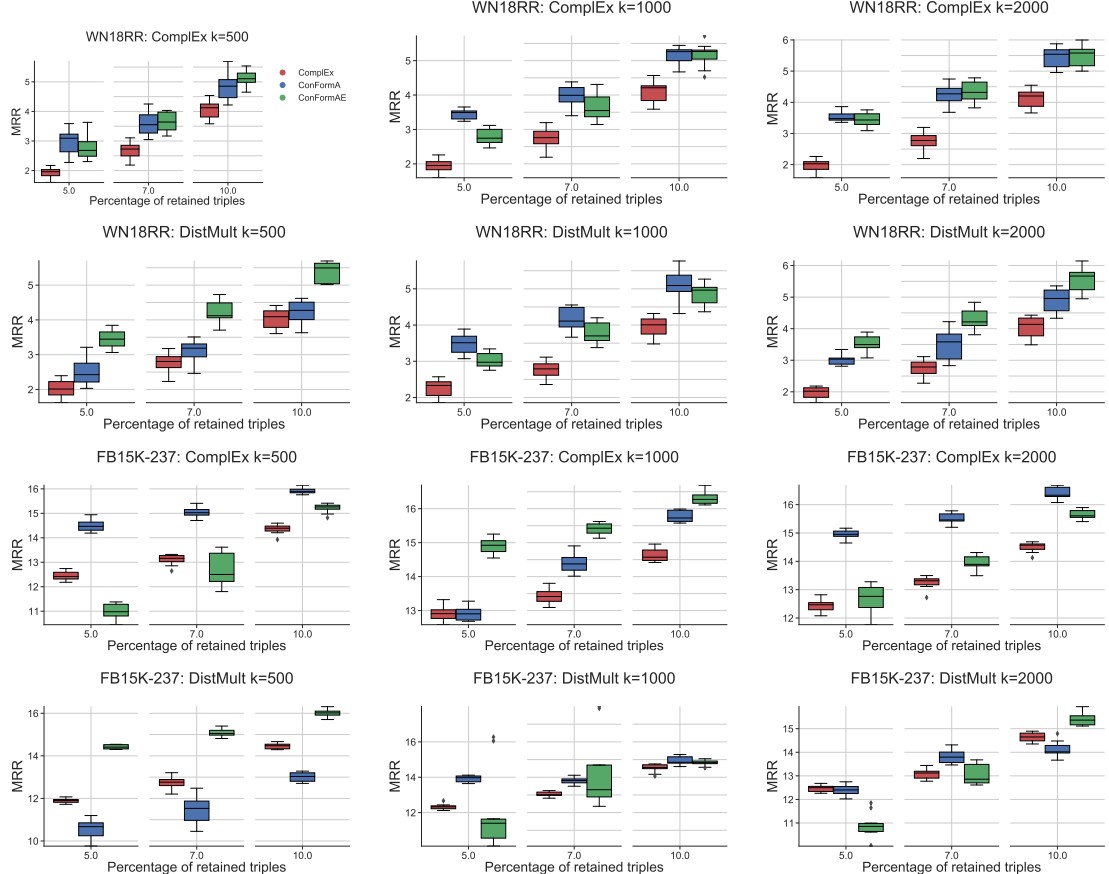

Table 25: MRR of ConFormA, ConFormAE, and baseline models – either DistMult or ComplEx – on sparsified WN18RR and FB15K-237, where the percentage of retained training triples is in $\{5, 7, 10\}$, for different values of embedding size ($k$). Each experiment was repeated with 6 different random seeds.

Table 26: Bins for categorizing relations into sub-populations based on their frequency, $N$, in the training set.

| Sub-population | WN18RR | FB15K-237 | UMLS |
|---|---|---|---|
| Rare | $N < 10^3$ | $N < 10^2$ | $N < 20$ |
| Medium | $10^3 < N \leq 10^4$ | $10^2 < N \leq 10^3$ | $20 < N \leq 150$ |
| Common | $N > 10^4$ | $N > 10^3$ | $N > 150$ |

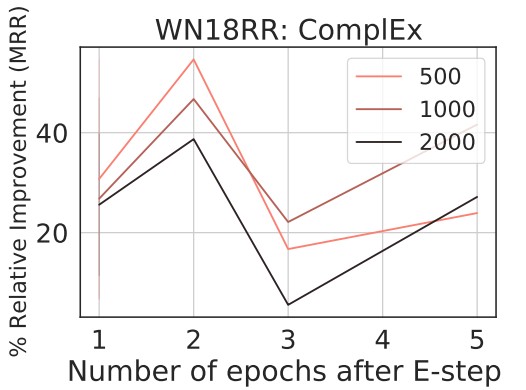 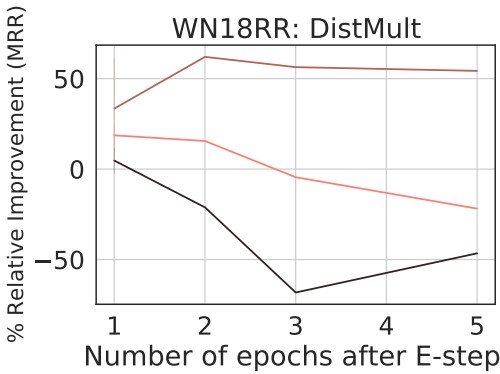

Figure 5: The relative improvement on Mean Reciprocal Rank (MRR) with varying the number of neural link predictor training epochs between every E-step in CONFOR-mAE, shown for WN18RR with ComplEx and DistMult, with rank size ($k$) in $\{500, 1000, 2000\}$. Each experiment was repeated with 5 different random seeds and averages plotted.

