# OpenReview forum: "Neural Concept Formation in Knowledge Graphs"
_AKBC.ws/2021/Conference — AKBC 2021_

### Official Review · Reviewer_FyiY · 2021-07-17
**Contributions not exactly novel**

**Rating:** 4
**Confidence:** 4

**Review:**

The authors propose two algorithm for unsupervised concept learning in knowledge graphs: ConFormA and ConFormAE, where the latter is iteratively optimizing the concepts. In their experiments, the authors investigate the semantic meaningfulness of the concepts (qualitatively), the effect on link prediction performance, the impact in low-data regimes, and the generalization ability for relation types of different frequency.

Strengths:
- The research questions are interesting and the experimental design is solid.

Weaknesses:
- The main weakness of the paper is that both contributions (1. task of unsupervised concept formation, 2. algorithms based on entity clustering and KG augmentation) are not novel but have been investigated before, e.g., in Gad-Elrab et al. (ISWC 2020, https://openreview.net/forum?id=wbVVXSziQFQ).
- In general, the related work section of the paper seems to be outdated as the latest cited paper in that section is from 2018.
- The meaningfulness of the concepts is evaluated only qualitatively by manually looking at some examples. This leaves room for questions about their actual quality. Although a large quantitative evaluation might not be possible due to missing gold data, there are other possibilities, such as user studies or manual annotations of a subset of clusters.

Other comments:
- Although the authors say that clustering KG embeddings did not lead to improvements, it would be valuable to add a comparison to this approach to the paper.
- Do the authors have an explanation why the joint training (i.e., ConFormAE) did not clearly outperform ConFormA? This is a bit unexpected. Could one reason be that the E and M steps are only loosely connected (in the form of the augmented graph but not in the form of the used embeddings for instance)? In general, I see only a loose connection of Algorithm 2 to "Expectation-Maximization" as the M-step optimizes a parameter set that is not used in the E-step.
- It is not clear how the number of clusters was chosen (neither the range nor how it was optimized) and which number was used in which experiments. Since it can be expected that this choice has a large impact on the performance, this would be very important to clarify.
- The introduction should also cite TransE as this was a link prediction approach clearly before the one that was cited.
- I don't follow the first reason why ConFormA should improve link prediction performance. Can the authors elaborate on that? Which knowledge does ConFormA add that is not implicitly contained in the embeddings for link prediction already? (This is even given as one reason for not using embeddings for clustering)

---

> ### Author Response · Authors · 2021-07-26
> **Response to Reviewer FyiY**
>
> Thank you for your valuable feedback and comments! Here we address your concerns.
>
> > Relationship with “ExCut: Explainable Embedding-Based Clustering over Knowledge Graphs”
>
> Thank you for sharing the “ExCut: Explainable Embedding-Based Clustering over Knowledge Graphs” (https://resources.mpi-inf.mpg.de/d5/excut/ExCut_TR.pdf) paper -- we were not aware of this work, it’s definitely related! We will make sure to incorporate it in the revised version of this paper.
>
> The first difference with Gad-Elrab’s work lies in our different objective: they aim at performing clustering for the sake of unsupervised explanations, whilst we focus on improving the accuracy of a downstream neural link prediction model. Our research question is whether representation learning models for Knowledge Graphs can learn better representations when such concept memberships are materialized in the Knowledge Graph.
>
> A similarity between our work and theirs lies in their iterative process for refining cluster memberships (Fig. 2 in the ExCut paper). This can be thought of as a particular instance of ConFormA where K-Means is used on latent embeddings and when the reification strategy of Ex-Cut materializes isA relationships. While we allow for other clustering routines, Ex-Cut allows for additional reification strategies, combining both could be an interesting future research direction. We will discuss it in the camera-ready version of this paper.
>
> Another difference can be found in our additional contribution, ConFormAE, as a way to overcome the choice of a particular clustering mechanism and as a principled probabilistic way to elicit entity concepts. There is no equivalent of the EM-based optimization loop of ConFormAE in Ex-Cut, which is heuristics in nature and lacks a clear optimization objective. We speculate that one could extend Ex-Cut to use a ConFormAE-like iterative scheme to learn clusters and explanations.
>
> > Experiments with ConFormA and KG embeddings: “do the authors have an explanation why the joint training (ConFormAE) did not clearly outperform ConFormA?”
>
> Thank you for your comments --  we added the experimental results for ConFormA with KG embeddings in the revised version of the paper (Appendix D, Table 9). Please see also our answer to this question in the joint answer we posted to all reviewers.
>
> > Relationship between Algorithm 2 and Expectation-Maximization: the M-step optimizes a parameter set that is not used in the E-step.
>
> In our framework, the E-step materializes the hard concept memberships, which depend on the current set of parameters (including embeddings for entities, clusters and ISA relationships). This is shown in the maximization problem we are solving over the predicted scores (please see line 7 of Algorithm 2). Then, the M-step updates all these parameters.
>
> > It is not clear how the number of clusters was chosen (neither the range nor how it was optimized) and which number was used in which experiments.
>
> All the hyperparameters, including the range for each dataset and how it was optimized (using the MRR on the held-out validation set), are specified in Appendix F.  Moreover, the impact of varying the number of clusters on MRR for both WN18RR and FB15K-237 has been investigated and reported in Figure 4.
>
>
> > The introduction should also cite TransE as this was a link prediction approach clearly before the one that was cited.
>
> Thank you for pointing this out, indeed a few neural link prediction models are not being cited in the paper, while we referred to the recent survey of Nickel et al [1]. We addressed this issue in the revised version.
>
> > Which knowledge does ConFormA add that is not implicitly contained in the embeddings for link prediction already?
>
> Clustering via random paths in ConFormA adds long-range information that is generally not captured by latent embeddings. This has been already observed in  [1] where authors state that the “strengths of latent and graph-based models are often complementary.”
> In fact, in our experiments, we show that, when concept memberships are initialized with graph-based features of the entities, such as their membership to paths, this can significantly improve the accuracy of downstream neural link prediction models.
>
> [1] Nickel, M. et al. - A Review of Relational Machine Learning for Knowledge Graphs. Proc. IEEE 104(1): 11-33 (2016) - https://arxiv.org/abs/1503.00759

---

> ### Author Response · Authors · 2021-07-31
> **More quantitative results**
>
> Thanks to the suggestion of Reviewer ygGy, we were able to provide an additional quantitative study over the quality of clusters learned by ConFormA/E.
> In a nutshell: ConFormA and ConFormAE deliver better clusters than directly clustering neural link predictor embeddings according to normalized mutual information scores and for different embedding sizes, base neural link prediction (ComplEx, DistMult) and across initializations with different number of clusters. Please see our updated revision, Appendix G, as well as our answers to other reviewers.

---

### Official Review · Reviewer_d4i2 · 2021-07-18
**Well-written paper, but more analyses on clustering quality would be nice**

**Rating:** 6
**Confidence:** 3

**Review:**

The paper aims to improve the performance of neural link predictors on sparse KGs, by augmenting KGs with clusters of concepts. It proposes two versions of augmentations: (1) ConFormA which inserts the clustered concept, and (2) ConformAE which jointly learn the concepts and embeddings through an EM scheme. The authors show that the augmentation helps improve performance of two baselines. on different datasets.

Strength:

S1. The research topic is interesting.

S2. The paper has clear motivations for each section. I especially like Q1-Q4 set at the beginning of the experiment.

S3. The paper explains its results nicely, with hypotheses/explanations for all the phenomena.

S4. The paper provides detailed appendices with additional experiment setups and results which are important for reproducibility.

Weaknesses / questions:

W1. The quality of clusters.

From the method description, It looks like the augmentation quality is greatly impacted by the clustering quality, and so I would appreciate it if the authors can offer more detailed analyses on the clustering results. For example, would the clustering also work less stably for more sparse concepts? Are their ambiguous entity that can belong to more than one concept which may cause the hard clustering to backfire?

I understand it might be difficult to do scaled quantitative analysis, but on top of the (nice) qualitative tables in the paper, I wish to at least see some more rigorous manual inspection. For example, if we manually inspect 20 concepts, in the top-10/ last-10 clusters, what's the ratio of correctly clustered entities? Are the sparse entities correctly clustered? Are there some quality difference between concepts from ConFormA and ConFormAE?

W2. Augmentation method.

I'm curious why the authors only choose to add ISA relationships. If the concept C for an entity E1 is indeed correct, would it work better/worse if the we also extended links between (E1,E2) to (C, E2)?

W3. (minor) the paper content can be better organized.

e.g. Table 1/2 are referred in the text after Table 3, and so it would also be nice if they occur closer to when they are explained in the text.

---

> ### Author Response · Authors · 2021-07-26
> **Response to Reviewer d4i2**
>
> We would like to thank the reviewer for their valuable comments and suggestions. We are glad that the reviewer finds the work interesting and our hypotheses and experiments to be well-explained and foster reproducibility. Below we address the points raised by the reviewer.
>
> **W1**: Quality of Clusters
>
> We concur with the reviewer’s concern that it is important to examine cluster quality in a quantitative manner - the primary challenge we have encountered in attempting to do this revolves around the lack of availability of ground-truth cluster information for large KGs, such as WN18RR or FB15K-237. The sheer scale of these KGs renders manual analysis challenging, as cluster size can range from a few dozen to over a thousand entities.
> > “For example, if we manually inspect 20 concepts, in the top-10/ last-10 clusters, what's the ratio of correctly clustered entities? Are the sparse entities correctly clustered?”
>
> Given that we do not have access to ground-truth information of cluster assignments, how does the reviewer suggest we define ‘top-10, last-10’ clusters? By showing whole clusterings as results, we demand this qualitative judgment from the reader.
>
> > “Are there some quality differences between concepts from ConFormA and ConFormAE?”.
>
> How does the reviewer propose we do this? In Table 8 we have provided the entire clustering of UMLS for ConFormAE - we have now also included the entire clustering of UMLS using ConFormAE (Appendix C, Table 9), with random paths representations. It is worth noting that the number of clusters in ConFormAE changes dynamically during training, while in ConFormA the number of clusters stays constant - this renders comparing the two clusterings challenging.
>
> > “Are there ambiguous entities that can belong to more than one concept which may cause the hard clustering to backfire?”
>
>  The focus of this work is on hard clustering, extending it to soft-clustering would require a soft-augmentation scheme, which is definitely an interesting avenue for future research. If an entity naturally belongs to multiple clusters, a good hard clustering would select only one of them, the most likely. While some information is lost, it is hard to see how this could “backfire”.
>
> **W2:** Augmentation Method
> >  “If the concept C for an entity E1 is indeed correct, would it work better/worse if we also extended links between (E1, E2) to (C, E2)?”
>
> We tinkered with this possibility but realized that concept augmentations are meaningful depending on the type of relationship between entities. Consider for instance the relationship ‘capital_Of’. For  a triple like (Rome, capital_Of, Italy), and (Rome, is_A, City) being the reified concept triple -- adding a triple (City, capital_Of, Italy) would be meaningless. Equivalently, we could not automatically add (Italy, is_A, City).
> Devising how and when a richer automatic relation-based augmentation scheme can be obtained is an interesting future research direction.
>
> **W3.** We thank the reviewer for suggesting improving the order of the tables to improve readability, we have updated the current version accordingly.

---

> ### Author Response · Authors · 2021-07-31
> **Additional Quantitative Results on Cluster Quality**
>
> As you suggested, we are providing a quantitative analysis that should help understand in a more rigorous way how good the clusters learned by ConFormA/E are.
> Please see our most recent global answer and our updated revision (Appendix G) for details.

---

### Official Review · Reviewer_ygGy · 2021-07-21
**Well-written paper with interesting idea but the evaluation could be strengthened**

**Rating:** 6
**Confidence:** 4

**Review:**

This paper studies the problem of knowledge base (KB) reasoning and propose to learn new concepts via clustering vectorized (propositionalized) entities and use the resulted entity-concept membership information as data augmentation for neural link prediction models.Two methods of finding such concepts and entity-concept memberships are proposed, one based on hard clustering of entity embeddings, and one based on an EM-style joint learning with the base neural link prediction model. Experiments are mainly conducted on FB15K-237 and WN18RR, which shows that the proposed methods can improve over ComplEx and DistMult, especially when sparsity is high.

Strengths:

- The paper is well-written and pretty easy to follow
- The proposed concept formation idea is interesting and plausible
- Some of the experiment designs are well thought-out and the results seem supportive

Weaknesses:

- I have reservations about the whole discussion around the concepts in existing knowledge graphs/bases. Concepts (or types/classes) are a central part of many KBs/KGs. I find it surprising that for the lack of quantitative intrinsic evaluation of clustering quality, it's claimed that "to the best of our knowledge there exists no large KG for which such information is available". Many KGs have a type system. For example, Freebase, of which FB15K-237 is a subset, has a strong type system. There's polymorphism and one entity could belong to multiple types, which is inconsistent with the hard partition assumption of the proposed methods (that is a strong assumption that could be easily violated), but I think one could at least use the Notable_Types in Freebase to do a reasonable large-scale quantitative evaluation of clustering quality. I'm not sure if the Freebase types and entity-type memberships are retained in FB15K-237 or are pruned in the downsampling process. If pruned, it will be interesting to see a comparison with the original Freebase type information added back to FB15K-237. Note that it's okay if the comparison is not favorable to the proposed method, because such type information is not always available for any KG, but simply omitting its existence is not okay.

- The improvement over baselines, while statistically significant, is somewhat marginal, and it's not clear whether the same improvement would translate to more advanced base models proposed more recently.

- It's not entirely clear to me why directly using embeddings learned by a neural link prediction model for clustering doesn't work well, and why using the method from Das et al [2020] would work instead. The provided explanation ("This could be due to the latent concept information which we explicitly introduce via augmentation being already captured by the neural link predictor embeddings") is hand-wavy.

---

> ### Author Response · Authors · 2021-07-26
> **Response to Reviewer ygGy**
>
> We would like to thank the reviewer for their insightful comments and suggestions, and for deeming our proposed concept formation framework interesting and experiments well thought-out.
>
> > If pruned, it will be interesting to see a comparison with the original Freebase type information added back to FB15k-237.
>
> Following the reviewer’s suggestion, we have tried to make this comparison but were unable to retrieve any type information for Freebase IDs from Wikidata or Freebase dumps.
>
> > The improvement over baselines, while statistically significant, is somewhat marginal, and it's not clear whether the same improvement would translate to more advanced base models proposed more recently.
>
> First, we would like to point out that, when combined with modern training procedures (as we do), ComplEx continues to achieve state-of-the-art performance compared to more recent alternatives -- please refer to [1] for a recent overview.
>
> Second, we believe our performance gains are not marginal, especially when looked at under the light of the sparsification experiment, which is the main motivation to use concept information.
>
> Third, please also note that our gains are on the same order of magnitude as other widely adopted regularization methods for neural link predictors. For example, our approach (Table 4) is either on par or outperforms the gains obtained by the reciprocal relation augmentation proposed by Lacroix et al, 2018, which is ubiquitously used nowadays. (Furthermore, as we show in Table 4, in most cases, combining the two augmentations is possible and yields the largest improvement).
>
> [1] D. Ruffinelli et al. - You CAN Teach an Old Dog New Tricks! On Training Knowledge Graph Embeddings. ICLR 2020
>
> > It's not entirely clear to me why directly using entity embeddings doesn't work well, and why using the method from Das et al. [2020] would work instead.
>
> Thank you for your comment - please see our joint response and the revised paper where we added an empirical evaluation (Appendix D, Table 10) where we investigated performing ConFormA with clustering over ComplEx embeddings.

---

> > ### Comment · Reviewer_ygGy · 2021-07-27
> > **Follow-up Discussion**
> >
> > Are you using the `type.object.type` property to pull type information from the Freebase dump? You could also try `type.object.notable_type` which will give you a smaller set of notable types. Considering that FB15K was based on a rather old version of Freebase, it’s likely that entities are still identified using their `en.` IDs like `en.barack_obama`, which was later deprecated in Freebase and replaced with uniform machine IDs such as `m.xxxxxx`. So If you were trying to find type information using the `en.` ID of entities, you won’t be able to fetch any.
> >
> > Along the same line, I think the paper could be strengthened by adding a careful discussion of the ontology/concepts/type systems of existing knowledge graphs/bases. As I indicated, concepts are a core part of KGs/KBs. The fact that the commonly used KB embedding benchmarks don’t happen to include the type system (I’m not entirely sure but it seems so according to the authors) doesn’t mean that it doesn’t exist. As the proposed technique is claimed to be rooted in concepts, a careful discussion is necessary.

---

> > > ### Author Response · Authors · 2021-07-31
> > > **Freebase entity types, role of concepts in KGs**
> > >
> > > > Freebase type information
> > >
> > > We sincerely thank the reviewer for the detailed information on how to retrieve the entity types.  We have been able to extract the type property from the Freebase dump for every entity in FB15k-237, which has allowed us to perform a range of insightful quantitative and qualitative experiments. We perform a similar analysis on UMLS for which we have also been able to extract ‘semantic group’ information.
> > > In Appendix G of the updated paper, we provide a quantitative analysis (Normalized Mutual Information and Jaccard Index) of the clusterings learned by ConFormA and ConFormAE against the gold clusters coming from the retrieved types (Freebase) or semantic groups (UMLS)  and compare against the other baseline that reviewers mentioned in their comments: directly clustering neural link predictor embeddings.
> > >
> > > Here we summarize our findings (see also updated paper):
> > > For UMLS, we find that ConFormA with random paths and ConFormAE outperform clustering neural link predictor embeddings and recover the group information rather faithfully, with ConFormAE achieving the best scores overall.
> > > For FB15k-237, we find that across all 3 approaches the NMI is relatively low, indicating that all the methods struggle to recover the concepts as specified by types.
> > > ConFormA with relatively low-dimensional random paths representations (d=474) consistently outperforms clustering neural link predictor embeddings, even of significantly larger dimensions.
> > > ConFormAE initialized with random paths representations improves over all other approaches.
> > >
> > > > Discussion of the ontology/concepts/type systems of existing knowledge graphs/bases.
> > >
> > > Thank you for this suggestion as well: in the revised version we discuss how concepts are first-class citizens in the knowledge representation formalisms used by Knowledge Graphs, like RDF and RDFS.

---

### Author Response · Authors · 2021-07-26
**Response to All Reviewers**

We would like to thank all the reviewers for their insightful suggestions and comments. In addition to more detailed individual replies, we clarify some common themes here.


**ConFormA: clustering entity embeddings learned by neural link predictors**

Reviewers FyiY and ygGy both inquired about empirical evidence regarding the statement that performing ConFormA via directly clustering entity embeddings does not result in performance improvements in downstream link prediction tasks, and the reasons why this is the case.
First, we would like to make it clear that it is not that clusters learned directly on latent embeddings are not meaningful but rather that they are not improving the accuracy of the neural link predictor -- which is the key objective of our work. In fact, the concept-membership information contained in the embedding representations is already implicitly exploited by the neural link predictor, and reifying it adds little information to the model.

To demonstrate this, we have updated the paper (Appendix D, Table 10) to include an empirical investigation in which we augment both WN18RR and FB15K-237 with concepts by clustering ComplEx embeddings. We report MRR results for a range of embedding sizes and number of clusters and find that in most cases the augmentation either degrades the accuracy of the neural link predictor, or yields no significant improvements.

As for why performing ConFormA using random paths representations leads to an improvement, sometimes even surpassing ConFormAE, we point to [1] where authors already observed that the “strengths of latent and graph-based models are often complementary.” Using graph-based features rather than latent features for computing the concept membership can provide additional information, improving the prediction accuracy in downstream link prediction tasks.

[1] Nickel, M. et al. - A Review of Relational Machine Learning for Knowledge Graphs. Proc. IEEE 104(1): 11-33 (2016) - https://arxiv.org/abs/1503.00759

---

### Author Response · Authors · 2021-07-31
**Further quantitive analysis on cluster quality**

We provide in Appendix G of the last uploaded revision a quantitative analysis of the clusterings delivered by ConFormA and ConFormAE on UMLS (based on the semantic groups of [1]) and on FB15K-237 (based on the `notable_type` attributes of the FB dump as suggested by reviewer ygGy.

We evaluate the quality of clusterings by the Normalized Mutual Information  score (or its generalized version to deal with the overlapping clusters found in FB) between the clusterings learned by ConFormA and ConFormAE against the golden clusters and compare against the other baseline that reviewers mentioned in their comments: directly clustering neural link predictor embeddings.

See the updated revision for details. In summary: on UMLS, ConFormA and ConFormAE outperform clustering neural link predictor embeddings and recover the group information rather faithfully, with ConFormAE achieving the best scores overall. For FB15k-237, we find that across all 3 approaches the NMI is relatively low, indicating that all the methods struggle to recover the concepts as specified by types.

[1] Olivier Bodenreider and Alexa T. McCray.  Exploring semantic groups through visual approaches.J. Biomed. Informatics, 36(6):414–432, 2003

---

### Decision · Program_Chairs · 2021-08-17

**Decision:**

Accept

**Comment:**

This paper investigates the learning of novel concepts in KGs via unsupervised entity clustering used for the augmentation of edge prediction models. While reviewers had some questions as to the strength of the experiments (thoroughness of baselines, strength of results above baselines), the novelty of methods proposed, and explanations for results, overall reviewers found the paper well-written and interesting.